# Contribution of Model Organisms to Investigating the Far-Reaching Consequences of PRPP Metabolism on Human Health and Well-Being

**DOI:** 10.3390/cells11121909

**Published:** 2022-06-13

**Authors:** Eziuche A. Ugbogu, Lilian M. Schweizer, Michael Schweizer

**Affiliations:** 1School of Life Sciences, Heriot Watt University, Edinburgh EH14 4AS, UK; amadike.ugbogu@abiastateuniversity.edu.ng (E.A.U.); lilian.schweizer1@btinternet.com (L.M.S.); 2Institute of Biological Chemistry, Biophysics & Engineering (IB3), School of Engineering &Physical Sciences, Heriot Watt University, Edinburgh EH14 4AS, UK

**Keywords:** phosphoribosyl pyrophosphate synthetase (PRS/PRPS), eukaryotic model organisms, *Saccharomyces cerevisiae*, *Danio rerio*, human neuropathies: Arts syndrome, CMTX5, DFN2/DFNX1, ageing, cancer, cell signalling, CWI pathway

## Abstract

Phosphoribosyl pyrophosphate synthetase (PRS EC 2.7.6.1) is a rate-limiting enzyme that irreversibly catalyzes the formation of phosphoribosyl pyrophosphate (PRPP) from ribose-5-phosphate and adenosine triphosphate (ATP). This key metabolite is required for the synthesis of purine and pyrimidine nucleotides, the two aromatic amino acids histidine and tryptophan, the cofactors nicotinamide adenine dinucleotide (NAD^+^) and nicotinamide adenine dinucleotide phosphate (NADP^+^), all of which are essential for various life processes. Despite its ubiquity and essential nature across the plant and animal kingdoms, PRPP synthetase displays species-specific characteristics regarding the number of gene copies and architecture permitting interaction with other areas of cellular metabolism. The impact of mutated *PRS* genes in the model eukaryote *Saccharomyces cerevisiae* on cell signalling and metabolism may be relevant to the human neuropathies associated with *PRPS* mutations. Human *PRPS1* and *PRPS2* gene products are implicated in drug resistance associated with recurrent acute lymphoblastic leukaemia and progression of colorectal cancer and hepatocellular carcinoma. The investigation of PRPP metabolism in accepted model organisms, e.g., yeast and zebrafish, has the potential to reveal novel drug targets for treating at least some of the diseases, often characterized by overlapping symptoms, such as Arts syndrome and respiratory infections, and uncover the significance and relevance of human *PRPS* in disease diagnosis, management, and treatment.

## 1. Introduction

Phosphoribosyl pyrophosphate synthetase (PRS; ATP:D-ribose-5-phosphate pyrophosphotransferase, EC 2.7.6.1) is an essential rate-limiting enzyme that catalyzes the irreversible transformation of ribose-5-phosphate in the presence of ATP to produce 5-phospho-D-ribosyl-α-1-pyrophosphate (PRPP) and AMP and is subjected to feedback inhibition by AMP, ADP and GDP [1,2,3,4,5]. PRS links carbon and nitrogen in cellular metabolism producing the high-energy compound PRPP, a key metabolite necessary for the biosynthesis of purine and pyrimidine nucleotides, the cofactors nicotinamide adenine dinucleotide (NAD^+^), nicotinamide adenine dinucleotide phosphate (NADP^+^) and, in specific organisms, e.g., *Escherichia coli*, the amino acids histidine and tryptophan [6,7]. PRS is responsible not only for the *de novo* synthesis of purine nucleotides but also plays a role in the salvage pathways *via* dedicated phosphoribosyl transferases for adenine, guanine and hypoxanthine. However, pyrimidine nucleotides, i.e., thymine and uridine, are salvaged by recycling from degradation products of DNA and RNA [8,9,10,11].

Synthesis of PRPP is essential for all free-living organisms but is lacking in parasites, e.g., *Trypanosoma* and *Leishmania*, which are dependent on the host metabolism for the synthesis of PRPP [12]. PRS enzymes have been identified in numerous organisms including mammals, human and rats [3,13,14,15,16,17]; plants, including *Spinacia oleracea* [18] and *Arabidopsis thaliana* [19,20]; bacteria, including *Bacillus subtilis* [21], and *Bacillus pseudomallei* [22]; yeasts, including *Saccharomyces cerevisiae* [23,24,25,26] and *Schizosaccharomyces pombe* [27]; and more recently, in zebrafish *Danio rerio* [28,29]; confirming the ubiquity of *PRS* genes across the five plant and animal kingdoms. There is now a plethora of *PRS* genes available for investigating the ramifications of altered PRPP synthetase on the physiology of various organisms (Figure 1).

Interestingly, the number of *PRS* genes in the genome varies from organism to organism. For instance, humans (HGNC PRPS), *S. pombe* (PomBase) and *Aspergillus nidulans* (AspGD) each possess three *PRS* genes, whereas *S. oleracea* (Spinach Base) and *Eremothecium* (*Ashbya gossypii* (Ashbya Genome Database)) genomes each contain four *PRS* genes. However, *S. cerevisiae* (Sacharomyces Genome Database (SGD)) and *A. thaliana* (AtGDB) harbour five *PRS* genes in their genomes. Irrespective of the number of *PRS* genes identified in different organisms, each organism must have at least a minimum of one gene coding for *PRS* to ensure survival, as is the case for bacteria [7,21,30]. An interesting discovery is that a gene, Hb*PRS4*, identified in the genome of the rubber tree plant *Hevea brasiliensis* Műll, a member of the family *Euphorbiacaea* [31,32], encodes a protein with 80% identity to Prs4 of *A. thaliana.* A BlastN study revealed a second sequence in the rubber tree genome with 84% similarity to the Hb*PRS4* gene. The expression of both genes responded positively to ethylene treatment, as did the ATP/ADP content of latex cells, thus linking PRPP synthesis to natural rubber production [33,34]. In a similar context, it has been reported that in an industrial yeast strain, the increased expression of the genes required for the synthesis of PRPP correlated with enhanced xylose utilization, emphasizing the significant role of PRPP for biofuel production from lignocellulose [35,36]. More specifically, it has been shown that overexpression of Sc*PRS3* increases the resistance of the yeast cell wall to acetic acid stress, thus illustrating the importance of Prs3 in the enhancement of commercially important pathways, e.g., biofuel production for lignocellulose, possibly as a result of improving cell wall strength [37,38]. *Mycobacterium tuberculosis,* the pathogen responsible for tuberculosis, contains a single *PRS* gene whose product is the only source of PRPP for a precursor of the bacterial cell wall and therefore is essential for the maintenance of cell integrity, thus providing a potential drug target in the treatment of tuberculosis [39,40,41,42].

Numerous scientific reports have shown that mutations in the *PRS* genes in different organisms cause various forms of cellular disruption, including neurological diseases and metabolic disorders. The nomenclature for *PRS* genes is species-specific: the *PRS* genes in humans are designated as h*PRPS*, of which there are three isoforms, h*PRPS1*, h*PRPS2* and h*PRPS3* (h*PRPS1L1*). The genes, h*PRPS1* and h*PRSP2*, are located on the long and short arms of the X-chromosome at positions Xq22.3 (OMIM 311850) and Xp22.2 (OMIM 311860) [43,44], respectively, whereas h*PRPS3* maps to chromosome 7 at position p21.1 (OMIM 611566) and is expressed specifically in the testis [15,17,45]. Missense mutations in the h*PRPS1* gene may lead to a gain-of-function associated with PRPS1 superactivity [46]. Other neurological disorders, such as Arts syndrome [47,48], syndromic and nonsyndromic sensorineural deafness (DFN2/DFNX-2) [49,50] and Charcot-Marie-Tooth disease type 5 (CMTX5) [51] are due to the loss-of-function in the h*PRPS1* gene. 

The sequence similarity of hPRPS proteins to those of the established eukaryotic model organisms *S. cerevisiae* (baker’s yeast) and *D. rerio* (zebrafish) has the potential to be instrumental in elucidating the ramifications of PRPP metabolism in human neuropathies. Biochemical and genetic studies have also identified the specific roles of h*PRPS1* and h*PRPS2* genes in one of the deadliest diseases, cancer. For instance, an h*PRPS2* knockout in experimental models has resulted in c-Myc-driven tumorigenesis [52,53], suggesting that h*PRPS2* may be a target protein for manipulating cancer cells, whereas it has been reported that a decrease in h*PRPS1* expression impairs the proliferation of tumour cells [54]. Both h*PRPS1* and h*PRPS2* genes have been identified in pluripotent stem cells, where they may contribute to increased stem-cell-associated biosynthetic capacity [55].

The ubiquity of PRS enzymes has allowed them to develop specific characteristics regarding the number of gene copies and acquisition of additional DNA supporting linkage to other aspects of cellular metabolism in different species. A comparison of the *PRS* genes and mutations thereof in model organisms has the potential to uncover the biochemical significance and relevance of PRS in disease diagnosis, management, and treatment. There are several user-friendly databases available for variants of *hPRPS* genes, e.g., the Leiden Open Variation Database (LOVD^3^), the Human Genomics Community (Varsome) and MalaCards, which provide further information on h*PRPS* genes and transcripts that associate them with diseases, thereby facilitating diagnosis. The research on *S. cerevisiae* PRS in relation to PRPP metabolism and its far-reaching influence on yeast as well as human physiology and well-being is the result of work carried out in the authors’ laboratory by a continuum of pre- and postdoctoral researchers. 

## 2. PRS-Encoding Genes in *Saccharomyces cerevisiae*

*S. cerevisiae*, a workhorse eukaryotic model organism which, in addition to its contribution to bread, wine, beer and Marmite^®^ production, has been exploited by biotechnologists for genetic and metabolic engineering by revamping and minimizing its genome through the ‘design-build-test-learn’ cycle of synthetic biology [56,57,58]. 

The whole yeast genome duplication that took place approximately 100 million years ago created a tetraploid yeast [59,60]. Over time, this unstable tetraploid yeast lost more than 80% of the duplicated gene copies. Nevertheless, of the remaining 6000 genes that make up the genome of the current species of *S. cerevisiae,* at least 10% are duplicated [61]. This implies that the remaining gene duplications must have made a positive contribution to the ancestral yeast’s relative fitness. Gene duplications provide a potential for increasing diversity, thereby permitting adaptation to new environments. If the products of the two gene copies are identical and autonomous, then one of the copies is available for alteration or can be lost. The alteration may provide a new link to the function of the other copy [59,62,63]. When the duplicated genes have interdependent functions, the chance of diversification is limited, but this can be overcome by retaining both the original and the altered copy since they are essential for the survival of the organism. Of the myriad gene products common to both humans and yeast, PRS in yeast is encoded by five unlinked paralogous genes as the result of genome duplication and subsequent evolution. The central role played by *PRS* gene products in cellular metabolism can be exploited to uncover potential therapeutic targets for PRPS-associated human diseases. 

### 2.1. The Role of NHR1-1 in the Provision of PRPP and Maintenance of CWI

The *S. cerevisiae* genome contains five Sc*PRS* genes, namely Sc*PRS1*, Sc*PRS2*, Sc*PRS3*, Sc*PRS4* and Sc*PRS5,* which are located on chromosomes XI, V, VIII, II, and XV, respectively [24,25,26]. Remarkably, Sc*PRS1* and Sc*PRS5* have acquired in-frame non-homologous regions (NHRs), which are not introns since they are still present in the mature polypeptides, as demonstrated by Western blotting with tailor-made antibodies [24,25,64]. The gene products Prs1 and Prs5 differ in length from Prs2 and Prs4 since they contain non-homologous regions NHR1-1, NHR5-1 and NHR5-2. In each instance, one of the component monomers of the essential minimal functional units, Prs1/Prs3, Prs2/Prs5 and Prs4/Prs5, contains an NHR. Both NHR1-1 with a length of 105 aa and NHR5-2 with 65 aa disrupt the catalytic flexible loops of Prs1 and Prs5 (Figure 2), respectively, rendering these polypeptides incapable of PRPP synthesis. NHR5-1 has a length of 113 aa and is located within the regulatory flexible loop and contains no less than nine predicted serine, threonine or tyrosine phosphorylation sites (PhosphoGrid and Phosphopep) [65,66]. A distinguishing feature of NHR1-1 and NHR5-1 is that they contain aa (amino acids) motifs associated with phosphorylation or ubiquitination [67]. S_199_ is located just prior to the beginning of the NHR1-1 sequence and has been postulated to play a role in mTORC1 signalling [68]. In the case of NHR5-1 S_183_, located centrally, it has been identified as being phosphorylated after rapamycin treatment [68], although there are other possible candidate aa residues in the near vicinity [69]. NHR5-2 is a stretch of 65 aa, including five serine residues, of which three, at the neighbouring positions S_364_, S_367_ and S_369_, are phosphorylatable [70]. Interestingly, characterization of a rapamycin-sensitive phosphoproteome revealed that Prs5 is hyperphosphorylated following rapamycin treatment [71,72], but the locations of the phosphosites were not specified. A high sequence similarity of 77% has been found for Prs1 between *S. cerevisiae* and *A. gossypii* (Ashbya Genome Database, AER083c) [73]. This similarity is maintained across the N-terminal sections of NHR1-1 in both species up to the midpoint of NHR1-1 of *S. cerevisiae* (unpublished data). Loss of the N-terminal portion of NHR1-1 in *S. cerevisiae* coincides with the inability to respond to heat shock as measured by Rlm1 expression, a transcriptional readout of the CWI (cell wall integrity) pathway [74]. NHR3-1 (Figure 2), close to the C-terminus of Prs3, is the pentameric sequence _284_KKCPK_288_, corresponding to an NLS (nuclear localization signal) consensus sequence of two positively charged aa (K (lysine) or R (arginine)) flanking three residues, one of which is P (proline), as described in [75].

A systematic phenotypic analysis revealed that individual deletion of any of the five *S. cerevisiae PRS* genes (Sc*PRS1*-Sc*PRS5*) did not seriously affect the cell’s viability [23,24,25,76]. However, *S. cerevisiae* cannot survive with a single *PRS* gene. Genetic evidence and Y2H (yeast-two-hybrid analysis) have shown that PRS exists as two interacting complexes: one as a heterodimer (Prs1/Prs3) and the other as a heterotrimer (Prs2/Prs4/Prs5). Furthermore, at least one of the three different combinatorial functional units, Prs1/Prs3, Prs2/Prs5 or Prs4/Prs5, must be present to ensure viability [26,76]. Expression of the five possible pairwise combinations of the Sc*PRS* gene products in *E. coli* achieved only 25% of the level of the *E. coli* PRPP synthetase with only the combination, Prs1/Prs3 [77] which correlates with Prs1/Prs3 being the most important heterodimer for yeast survival [25]. Mutation of Sc*PRS1*, Sc*PRS3* and Sc*PRS5* have proven useful for investigating the link between Prs and its specific functions—nucleotide biosynthesis, cell signalling and metabolism [26,76,78,79,80]. Overexpression of Pkc1 had a positive effect on the transcriptional activation of Rlm1 in either *prs1Δ* and *prs3Δ* strains at ambient and elevated temperatures and following α-factor stimulation [78].

A reduction in the capacity to synthesize PRPP, severe loss of enzymatic activity, compromised CWI, increased chitin content and severe growth retardation in *S. cerevisiae* were observed when Sc*PRS1*/Sc*PRS3* were simultaneously deleted in the absence of Sc*PRS2* or Sc*PRS4* (Figure 3a). Therefore, the Prs1/Prs3 heterodimer is the most important [25,74] since the mutant strains relying on either Prs2/Prs5 or Prs4/Prs5 are severely impaired in their growth, their PRPP-synthesizing capacities and CWI, thus providing genetic evidence to support the corresponding enzyme activities measured in yeast. Furthermore, yeast PRS enzyme activity was reduced to 80% when Sc*PRS2*, Sc*PRS4* and Sc*PRS5* were deleted simultaneously, but there was no influence on nucleotide content or growth rate (Figure 3b) [26,76,80]. Simultaneous deletion of Sc*PRS1* and Sc*PRS5* or Sc*PRS3* and Sc*PRS5* causes synthetic lethality, as does deletion of Sc*PRS2* and Sc*PRS4* in addition to the loss of Sc*PRS1* or Sc*PRS3* [26,76] (Figure 3c).

By performing a variety of diagnostic tests, including environmental exposure, e.g., heat shock, caffeine sensitivity, CFW (calcofluor white), Congo red and exposure to mating pheromone, we discovered that strains deleted for Sc*PRS1* or Sc*PRS3* are compromised in the maintenance of CWI [64,78,81,82]. The general architecture of the MAPK (mitogen-activating protein kinase) pathway in eukaryotes is illustrated in Figure 4a. Excellent reviews of the yeast CWI pathway and the extent of its influence on yeast physiology have been published [83,84,85,86,87,88,89,90,91]. Sc*PRS* mutant strains have consistently shown varying degrees of sensitivity to the purine analogue, caffeine (1,3,7-tri-methylxanthine), thus compromising CWI signalling *via* the MAPK (mitogen-activated protein kinase) [64,74,79,82,92]. It should be noted that caffeine not only interferes with the integrity of the cell wall, but also possibly by the inhibition of mTOR (mechanistic Target Of Rapamycin) mTORC1 and mTORC2, well-known regulators of eukaryotic cellular metabolism [93,94,95].

The spectrum of caffeine-sensitive phenotypes associated with Sc*PRS* deletants ranges from the growth of WT and Sc*PRS2* deletants at 6.5 mM caffeine dropping to growth at 5.5 mM caffeine for *prs4Δ* and 4.5 mM for *prs5Δ* strains. Deletion of Sc*PRS1* or Sc*PRS3* increases caffeine sensitivity to a concentration of 3 mM being tolerated, whereas a double deletant of *prs1Δ prs3Δ* fails to grow on 2 mM caffeine. The degree of caffeine sensitivity found in the double deletants *prs1Δ prs3Δ* and their inability to recover from exposure to caffeine, as demonstrated by the release of alkaline phosphatase, is further evidence of a link between Sc*PRS* gene products and the maintenance of CWI, an indication of cell disruption or cell alteration [64,82]. The increased sensitivity to caffeine can be reversed by the addition of 1 M sorbitol as an osmotic stabilizer. Additional hallmarks of impaired CWI, e.g., heat and cold sensitivity, altered chitin content, sensitivity to the yeast mating pheromone, α-factor and CFW resistance have also been observed in Sc*PRS* deletants (Figure 3a) [26,76,78,80,82]. Pkc1 controls the CWI MAPK cascade, [83,90] and we have shown that overexpression of yeast Pkc1 relieved α-factor sensitivity [78]. DIC (differential interference contrast) imaging of all Sc*PRS* deletants revealed an altered morphology in comparison to the WT. Specifically, *prs3Δ* strains are defined by highly vacuolated, large spherical cells, whereas *prs5Δ* cells have an elongated form, indicative of a defect in polarized growth. Interestingly, the triple deletion *prs1Δ prs3Δ prs4Δ* causes a higher incidence of plasma membrane invagination than the WT strain [64]. A strain deleted for one of the cell surface sensors, Mid2 (Figure 4b), thereby activating the CWI pathway, is also sensitive to α-factor, but the combination of a *mid2Δ* deletant with either a *prs1Δ* or *prs3Δ* deletant did not increase the α-factor sensitivity, suggesting that the Prs1 and Prs3 polypeptides interact with downstream components of this pathway [78,82]. Further evidence for the interaction of Prs polypeptides with the CWI pathway has been demonstrated in an extensive Y2H analysis revealing that Rlm1, an endpoint of the CWI pathway, interacts with Prs1, Prs3 and Prs5 (Figure 4b) [74,78,79,92]. Congo red, like caffeine, is a screening agent for yeast mutants with altered cell wall architecture [84,89]. Exposure of the collection of single Sc*PRS* deletants to Congo red revealed that only the *prs1Δ* strain did not recover after 3 days, which was in contrast to the remaining deletants. No recovery of a *prs1Δ prs3Δ* strain was observed, confirming the importance of the Prs1/Prs3 heterodimer for cell viability [81].

Following gene duplication, yeast can sacrifice sources of PRS enzyme activity caused by the insertions, e.g., NHR1-1 and NHR5-2, since survival depends on the association of two Prs polypeptides, one containing at least one NHR and the other without an NHR. A concrete example of this is the Prs1/Prs3 heterodimer, which is the most important of the three minimal functional units required for yeast viability [27,75]. The insertion NHR1-1 in Prs1 disturbs the catalytic flexible loop, thus increasing its distance from the ribose-5-phosphate loop (Figure 2) and impacting its enzyme properties negatively but still permitting an interaction of Prs1 with the components of the CWI pathway, *viz.*, the MAP kinase, Slt2/Mpk1 [92], a functional homologue of human ERK5 (extracellular signal-regulated kinase) [100,101], and Mkk1 (mitogen-activated protein protein-kinase) [78,81,102]. A comparable situation exists for Prs5, which, in combination with either Prs2 or Prs4, provides the two remaining minimal functional units essential for cell viability. The three insertions referred to above, explain the necessity of retaining at least one of the three minimal functional units, Prs1/Prs3, Prs2/Prs5 or Prs4/Prs5, since only then can the production of PRPP and the maintenance of CWI in the yeast cell be guaranteed. Furthermore, it also explains why simultaneous deletion of Sc*PRS1* and Sc*PRS5* or Sc*PRS3* and Sc*PRS5* is synthetically lethal since neither PRPP production nor maintenance of CWI is sustained (Figure 3c).

By Western blotting with human anti-phospho MAPK antibodies, which recognize *S. cerevisiae* phosphorylated Slt2, we demonstrated that strains lacking Sc*PRS2*, Sc*PRS4* or Sc*PRS5* can activate Slt2 in response to heat stress in a pattern similar to that of the WT. However, the deletion of *ScPRS1* or *ScPRS3* causes constitutively phosphorylated Slt2 in the absence of heat stress. In a *prs1Δ* strain, there was a strong signal at time zero, which, after heat shock, was maintained over a period of four hours before falling to zero. A *prs3Δ* strain can maintain and increase the level of phosphorylated Slt2 upon exposure to heat stress [78,80,81]. The difference in the pattern of heat-induced Slt2 phosphorylation response in a *prs3Δ* strain is, in fact, the response of a *prs1Δ prs3Δ*, since it has been demonstrated that deletion of Sc*PRS3* or only its _284_KKCPK_288_ motif, which we designated NHR3-1 (Figure 2), results in simultaneous loss of Prs1 [92]. The sudden disappearance of the phosphorylated form of Slt2 seen in the Sc*PRS1* at five hours of heat exposure suggested an inability of this strain to endure stress over a longer period, possibly due to the breakdown of the Prs1/Prs3 heterodimer [80,81]. 

We have shown by co-immunoprecipitation that NHR1-1 interacts with the MAP kinase, Slt2, of the CWI pathway only when Slt2 is phosphorylated, thus providing more evidence for linking PRPP synthesis to the maintenance of CWI in yeast. A strain expressing non-phosphorylatable Slt2 (slt2(T_190_A, Y_192_F)) does not interact with Prs1; however, the kinase-dead Slt2 mutant (slt2(K_54_R)) still interacts with Prs1 [79,103]. It has also been possible to demonstrate that Prs1 is bifunctional since a measurable increase in the strength of the Y2H interaction between prs1(ΔNHR1-1) and Prs3 results in a 50% increase in PRS activity in comparison to the WT. A lower, albeit possibly more accurate factor, 15%, is measured when comparing PRS activity with strains carrying plasmid-borne versions of WT Prs1 and prs1(ΔNHR1-1) (Figure 5a). The inability of a prs1(ΔNHR1-1) strain to increase Rlm1 expression in response to heat stress (Figure 5b) while maintaining enzyme activity (Figure 5a) indicates, for the first time, that the two metabolic functions—provision of PRPP and maintenance of CWI—are separable and the presence of NHR1-1 is not essential for the formation of the Prs1/Prs3 heterodimer, as demonstrated by a stronger Y2H interaction with Prs3 (Figure 5c) [92].

Interestingly, Prs1-Prs5 interact with Nuf2, a spindle pole protein required for chromosome segregation and spindle pole activity [82,92,104]. The interaction of Prs3 with Nuf2, a component of the kinetochore Ndc80 complex that is localized in the nucleus, provides further evidence that Prs1, Prs3 and Prs5 are at least temporarily in the nucleus, which is strengthened by the observation that Prs1 is found in the nuclear fraction [24,25,64].

An outer mitochondrial membrane protein, Atg32, required for mitophagy, has been found to interact negatively with Prs5, i.e., the combination of deleting Sc*PRS5* and Sc*ATG32* simultaneously results in a more severe fitness defect BioGrid^4.4^ (HTP (high-throughput) score −0.1312) [98] than when each is individually deleted, providing further evidence of the wide-reaching effects of PRPP synthesis in yeast physiology. Prs1 interacts physically with Esa1, the catalytic subunit of the histone acetyltransferase Nua4 complex, which is involved in cell cycle progression and epigenetic transcriptional activation of histones of the nucleosome and regulation of autophagy [105].

### 2.2. The Presence of NHR5-2 Influences Rlm1 Expression, the Phosphorylation Status of Slt2 and the Interaction with the Gsk3 Kinase Rim11

NHR5-2, like NHR1-1, located in the catalytic flexible loop of the prototype yeast Prs2 polypeptide (Figure 2), contains three neighbouring phosphosites pS_364_ pS_367_ pS_369_ (p = phosphorylated serine residues) [70]. Mutations of these phosphosites, individually or in all possible combinations, influenced the transcriptional readouts of Rlm1 and Fks2. The latter is a stress-induced alternative subunit of the 1,3-β-D-glucan synthase under the control of the CWI pathway. When all three phosphosites were mutated, there was elevated Rlm1 and Fks2 expression at ambient temperature, which was further increased, albeit to a lesser extent than the WT, following incubation at elevated temperature (Figure 6a,d) [79]. The triple mutant designated prs5(479), in which the serine residues of the three phosphosites pS_364_ pS_367_ pS_369_ have been mutated to alanine, caused hyperphosphorylation of Slt2 at ambient temperature (Figure 6b), thus implying that both NHR1-1 and NHR5-2 are instrumental in the maintenance of CWI [74,79].

The observed synthetic lethality of a *prs3Δ prs5Δ* strain may, in fact, be linked to Sc*PRS1* since deletion of Sc*PRS3* is accompanied by the loss of Sc*PRS1*. Therefore, a *prs3Δ prs5Δ* strain is, in fact, a triple deletant, *prs1Δ prs3Δ prs5Δ*, consistent with the inability of Prs2 and Prs4 to support viability, thus strengthening the argument that Prs5 contributes to CWI since the presence of Prs5 provides a link to CWI signalling in the two minimal functional units Prs2/Prs5 and Prs4/Prs5 [74].

We have preliminary evidence that the triply phosphorylated NHR5-2 is a target of Rim11, one of the four yeast homologues of glycogen synthase kinase-3 (Gsk3), since Y2H analysis has shown that Rim11 and Prs5 interact [81,106] and this interaction is dramatically reduced when the phosphosites pS_364_ pS_367_ pS_369_ (prs5(479)) are mutated (Figure 6c & unpublished data). A prerequisite priming site essential for Gsk3 phosphorylation [107] is found in NHR5-2 (SGD). Gsk3, originally found to be involved in the regulation of glycogen synthase, is known to have multiple substrates, and is considered to play a role in various biochemical pathways, e.g., signal transduction, cellular metabolism, and neurodegenerative diseases [108]. These data provide further evidence that the insertions in Sc*PRS1* and Sc*PRS5* are not gratuitous but have probably arisen following the whole genome duplication predating the present *S. cerevisiae* genome. 

### 2.3. The Role of NHR3-1, the _284_KKCPK_288_ Motif in Prs3

The pleiotropic effect observed when Sc*PRS3* is deleted is a strong indication that Prs3 is essential for CWI, cellular homeostasis, and organization of the actin cytoskeleton in *S. cerevisiae* [25,74,76,79,92,109,110]. The pentameric motif NHR3-1, _284_KKCPK_288_ of Prs3 (Figure 2), is deemed necessary for intracellular transport since it is found in proteins which locate to the nucleus [75]. It was observed that the deletion of this pentameric motif destabilizes the Prs1/Prs3 heterodimer complex, as shown by Western blotting, increased caffeine sensitivity and a drastic reduction in Rlm1 expression at ambient and elevated temperatures as well as a reduction in PRPP synthetase activity, thereby mimicking the genotype *prs1Δ prs3Δ prs5Δ* [92], which is incompatible with the existence of any of the three minimal functional units required for cell viability. We postulate that Prs3 may be a specific transport protein linking PRPP synthesis to CWI by bringing the complex to the nucleus due to the presence of _284_KKCPK_288_. The _284_KKCPK_288_ motif of Prs3 is not present in any of the remaining four Prs polypeptides. This is corroborative evidence that a lack of Sc*PRS3* or merely the removal of NHR3-1 results in the loss of Sc*PRS1* [92]. Given the interaction of Prs1 with Mkk1 [80,102], and Slt2 [78,79], together with the interaction of Prs1, Prs3 and Prs5 with Rlm1 [92] (Figure 4b), it could well be that Prs3 facilitates the entry of Slt2 into the nucleus under conditions of stress since Slt2 may be located in both the nucleus and the cytoplasm [84,111,112]. Prs3 may be considered to both stabilize the Prs1/Prs3 minimal functional unit and ensure that under conditions of stress, together with Prs5, it enters the nucleus, where the transient four-component complex Prs1/Prs3/*P*-Prs5/*P*-Slt2 interacts with Rlm1 to initiate the stress response. Furthermore, the deletion of _284_KKCPK_288_ prevents rescue of the synthetic lethality of a *prs3Δ prs5Δ* strain, indicative that NHR3-1 is essential for viability by maintaining the existence of the Prs1/Prs3 heterodimer [92].

### 2.4. Prs1 and Prs5 Influence Fks2 Expression

The expression of Fks2, a catalytic subunit of the 1,3-β-D-glucan synthase, is, in comparison to Fks1, increased in response to heat stress and is regulated not through the G1-specific cell cycle but in response to cell wall stress [83,84]. This temperature-induced Fks2 response is reduced by more than 50% following the loss of Sc*PRS1*, Sc*PRS3* or Sc*PRS5*. A deletion of Sc*PRS5* almost completely abolishes the temperature-induced expression of Fks2 and correlates with Rlm1 expression in the same background [78]. This prompted us to examine the influence of the three phosphosites in NHR5-2 on Fks2 expression. When mutated individually, each of the mutations caused an increase in expression of Fks2 at 37 °C, albeit decreased, in comparison to the WT [79]. However, the triple mutant prs5(479) showed the highest response of Fks2 expression at ambient temperature [79] (Figure 6d) and a more modest increase at 37 °C. A similar temperature-dependent response has been documented for Rlm1 expression in the same strains (Figure 6a). These results link the phosphorylation sites in NHR5-2 to Rlm1 and Fks2 transcriptional readouts to the phosphorylation status of Slt2 [79]. Fks2 expression is dependent on Swi4/Swi6, also known as SBF, since phosphorylated Slt2 or Mlp1, the pseudokinase paralogue of Slt2/Mpk1 [83,84], interacts with Swi4 to bind to the Fks2 promoter (cf. Figure 4b). Recruitment of Swi6 to this complex permits the assembly of RNA polymerase II and the Paf1 elongation complex [97,113,114,115,116,117]. In the absence of Paf1, Prs1 transcription is reduced by 50% [114], thereby mimicking the deletion of Sc*PRS1,* which is associated with a reduction of Fks2 transcription [78,80]; this is a scenario leading to compromised CWI, consistent with the phenotypes associated with perturbation of the Prs complexes [74,76,78,79,80,92]. Furthermore, disruption of Sc*PAF1* negatively impacts on the induction of galactose-regulated genes [113]. Strains lacking Sc*PRS1* or Sc*PRS3* are unable to grow on galactose, whereas the deletion of Sc*PRS2*, Sc*PRS4* and Sc*PRS5* has no effect on galactose utilization, indicating that the Prs1/Prs3 and Prs2/Prs4/Prs5 complexes have different responsibilities in yeast metabolism [unpublished data]. 

### 2.5. Yeast Genocopies of PRPS-Associated Human Neuropathies Interfere with CWI

Prs2, Prs3 and Prs4 of yeast and hPRPS1 proteins share a sequence similarity of approximately 70%, which drops to 53% in Prs1 on account of the presence of NHR1-1 (Clustal Omega). Genocopies of mutations in Sc*PRS1* associated with human neuropathies were created to examine their impact on yeast physiology. Missense mutations in the h*PRPS1* gene can lead to a gain-of-function associated with PRPS1 superactivity [118,119,120,121] or a loss-of-function [51,118]. Arts syndrome [47,48] and Charcot-Marie-Tooth disease-X5 (CMTX5) belong to the latter category [51]. Genocopies in Sc*PRS1* of CMTX5 (c.343C > A + c.344T > C (p.L115T) (cf. p.M115T (c.344T > C) in h*PRPS1*) and Arts syndrome (c.398A > C (p.Q133P) (cf. p.Q133P (c.398A > C) in h*PRPS1*) were created. These genocopies are located at 86 and 68 aa, respectively upstream of NHR1-1 in Sc*PRS1*. Furthermore, mutations of the divalent-cation binding site alone or in combination with the ribose-5-phosphate binding site were created [74]. All constructs tested retain NHR1-1 and can increase Rlm1 expression at 37 °C, albeit to varying degrees (Figure 7a). Rlm1 expression in the presence of the genocopies p.L115T and p.Q133P was increased with respect to the WT at ambient temperature, indicating that these mutations may influence the folding of Prs1, which does not interfere with NHR1-1/Slt2 interaction [74]. A similar response of Rlm1 expression was observed when we tested the effect of mutating the divalent cation-binding site at position H130A (c.388C > G + c.389A > C) and the ribose-5-phosphate binding site, D326A (c.977G > C), singly and in combination, namely, an increased Rlm1 response at 30 °C, which increased further following incubation at 37 °C, suggesting that the folding of the mutated Prs1 polypeptides is altered in such a way that NHR1-1 can interact more readily with Slt2, possibly on account of their locations external to each monomer of the hexameric complex of *B. subtilis* PRPP synthetase complex [21]. Both NHR1-1 and p.H130A are essential for overcoming the synthetic lethality of a *prs1Δ prs5Δ* strain, whereas Prs1 with an altered ribose-5-phosphate binding site, p.D326A, can rescue the synthetic lethality [74,103]. The mutations p.H130A and p.Q133P are separated by only two aa but have significantly different levels of Rlm1 expression at 30 °C, suggesting that the helix breaker mutation, p.Q133P, has a more profound effect on protein folding. The Rlm1 expression data follow the same pattern as PRS activity (Figure 7b) [74]. Strains carrying these genocopies could suppress the synthetic lethality of the double mutant *prs1Δ prs5Δ* at 30 °C but did not restore growth at 37 °C. However, the inclusion of 1 M sorbitol as an osmotic stabilizer permitted growth at 37 °C, except for the strain carrying the genocopy of the mutation p.Q133P associated with Arts syndrome. Examining these rescued strains for their caffeine sensitivity indicated that in the absence of an osmotic stabilizer, growth was possible at 30 °C in 2 mM caffeine, although again, the genocopy p.Q133P (Arts syndrome) grew less well than the others. When 1 M sorbitol was included in the media, growth was restored at 30 °C at a concentration of 3 mM caffeine [103].

Molecular dynamics (MD) simulation of hPRPS1, p.Q133P, showed that there are fewer contacts between Pro133 and the catalytic flexible loop due to the lack of hydrogen bonds, which may explain the profound effect of this mutation, characteristic of Arts syndrome in h*PRPS1* [47,48]. It should be noted in h*PRPS1* that the mutation p.H130A (c.388C > G + c.389A > C) does not display normal AMP activity but shows an ADP activity with an almost two-fold increase in K_M_ and a ⅓ reduction in kinetic efficiency. Although the crystal structures of these mutants closely resemble that of the WT, these structural changes impact the intrinsic interactions of the enzyme, which influences activity rather than structure [122]. Such single aa changes have the potential for CRISPR-mediated therapy or drug development.

## 3. Dr*PRPS1* in the Model Organism *Danio rerio*

In zebrafish, there are only two *PRPS1* paralogues: Dr*prps1a* and Dr*prps1b* (ZFIN). The Dr*prps1a* gene is located on chromosome 5, whereas Dr*prps1b* is located on chromosome 14 [29]. Sequence alignment of hPRPS1 with zebrafish Prps1a and Prps1b revealed that Prps1a and Prps1b have a sequence similarity of 99% with hPRPS1 [28] (EMBOSS Needle). The mutant Dr*prps1a^la015591^* was created *via* retroviral vector insertion [123], whereas the Dr*prps1b^hg19^* mutant was engineered by zinc finger nuclease gene-editing by targeting exon 2, resulting in three frameshift mutations (*prps1b^hg19^*, *prps1b^hg20^* and *prps1b^hg21^*) [29]. The mutants of Dr*prps1a* were created using CRISPR/Cas9 gene-editing technology [29]. Genetic analysis of the generated mutants showed that both Dr*prps1a* and Dr*prps1a prps1b* double mutants had developmental abnormalities, such as neuromast hair, smaller eyes, optic atrophy, and defective hearing (cf. Figure 8a). The mutants exhibited a reduction in nucleotide synthesis and energy production that resulted in a prolonged cell cycle. The double mutant had reduced leukocytes, abnormal primary motor neurons and hair cell innervation development, whereas Dr*prps1b* mutants did not show any developmental abnormalities [29]. Begovich et al. (2020) [124] suggested that the mutation of Dr*prps1a* is sufficient to cause smaller eyes and defects in pigments, head size and swim bladder inflation. They also observed that Dr*prps1a* mutations lead to failure in assembling PRPS1 and disorders of actin networks in lens fibres and deduced that Drprps1a is the major contributor during normal embryonic development in zebrafish. The study of DeSmidt et al. (2019) [29] evaluated the auditory role of zebrafish prps1a and prps1b as a model to study hPRPS1 nonsyndromic X-linked sensorineural deafness (DFNX1/DFN2). They observed that the knockout of zebrafish Dr*prps1a* and Dr*prps1b* resulted in smaller otic vesicles, otoliths, and loss of inner ear cells, especially hair cells, in mutant strains, indicating that deletion of either of Dr*prps1a* or Dr*prps1b* results in sensorineural hearing impairment. It is not surprising that the mutant phenotypes observed in zebrafish Dr*prps1a* and Dr*prps1a* Dr*prps1b* double mutants are observed three dpf (days post-fertilization) but not later in development since it is known that zebrafish continue to grow over their lifetime rather than having specific developmental stages, which weakens the argument for using zebrafish as a model organism for PRPS deficiency-associated human diseases. Nevertheless, the sequence similarity in zebrafish and human PRPS is evidence of the fundamental role of PRPP in providing the building blocks of purines and pyrimidines. Depending on whether the Dr*prps1a* mutation causes either a reduction in mRNA expression [28] or displays pleiotropic effects arising from a truncated version of the protein [124], both lead to smaller eyes. The truncated version is also responsible for defects in head size, pigmentation, and swim bladder inflation consistent with Dr*prps1a* being the primary paralogue. Furthermore, since the truncation of Drprps1a removes most contact sites between the enzyme subunits, it would suggest that not only PRPS enzyme activity but also the PRPS filament assembly is absent, leading to a disorganized actin network in the lens fibres, which is only visible at a later stage of development. These observations are further evidence that Dr*prps1a* is the dominant allele in zebrafish. 

The reliance on enzyme activity and enzyme polymerization could be the reason for the various PRPS-associated disease phenotypes. For instance, gain-of-function mutations cause gout by overproduction of purines and hPRPS enzyme activity. However, mutations that disrupt feedback inhibition, also resulting in increased hPRPS activity, causes phenotypes associated with both loss-of-function and gain-of-function, e.g., sensorineural deafness and gout, respectively [50,125]. The spectrum of mutations associated with mutations in the *PRPS* genes may not only be the result of alterations of enzyme activity but could be due to the polymerization of PRPS, which, in zebrafish, influences actin fibre association in the lens, but which is visible only at a later stage of development. It has also been proposed that filament formation in PRPS could impinge on filament formation in other enzymes in the vicinity. It has been demonstrated using a GFP-labelled strain collection that specific gene products are capable of self-assembly, and Prs3 and Prs5 in *S. cerevisiae* form such composite structures [126]. As we have shown, all minimal functional units of PRPP synthetase contain either Prs3 or Prs5] [26,76]. The polymerization of individual Sc*PRS* gene products may have a regulatory function dependent on growth conditions. Transfer from a stationary phase into fresh media resulted in disassembly of the Prs3 and Prs5 filaments without any apparent loss of protein. The assembly/disassembly scenario is different in Prs3 from Prs5 in that Prs3 assembles only in the stationary phase, whereas Prs5 assembles in response to both acute glucose or adenine limitation and the stationary phase [126]. Such a situation is consistent with the concept of substrate availability controlling enzyme activity since the conversion of glucose to ribose-5-phosphate occurs *via* the pentose phosphate pathway. However, the necessity of at least one of the three heterodimeric functional units for the survival of the yeast cell would suggest the involvement of Prs1, Prs2 or Prs4 is essential for additional aspects of yeast metabolism. It must be noted that the Sc*PRS3* gene contains a sequence, NHR3-1, required for the stability of the Prs1/Prs3 heterodimer [92]. Furthermore, a contribution to the maintenance of CWI is integral to yeast PRPP synthetase since, during evolution following the diploidization of the ancestral yeast genome, the acquisition of genetic information NHR1-1, NHR5-1 and NHR5-2 have become essential for cell survival and may explain the different architecture of PRPP-synthesizing machinery in various organisms. This has been demonstrated by the failure of hPRPS1 to rescue the synthetic lethality of a *prs1Δ prs5Δ* yeast strain by plasmid shuffling [74]. Despite the structural organization of PRS in unicellular and multicellular eukaryotic organisms, the phenomenon of their filament formation has been conserved throughout evolution, as demonstrated by immunostaining in *Drosophila* egg chambers, rat neurons and human fibroblasts [126,127].

## 4. Human PRPS and Associated Disorders

In humans, the three isoforms of PRS, hPRPS1 (HGNC 9462), hPRPS2 (HGNC 9465) and hPRPS3 (HGNC 9463) [15,44,128,129,130,131,132], share high aa sequence similarity, namely 98.7% between hPRPS1 and hPRPS2, 97.2% between hPRPS1 and hPRPS3 and 96.5% between hPRPS2 and hPRPS3 (EMBOSS Needle). Human *PRPS1* exists in two isoforms. The PRPS1 transcript variant 1 mRNA, RefSeq NM_002764 contains seven exons and gives rise to a protein RefSeq NP_002755 with a MW of 34.8 kDa. The h*PRPS1* transcript variant 2 mRNA, RefSeq NM_001204402 gives rise to a shorter protein variant identical to the last 114 aa of PRPS1 isoform 1 (RefSeq NP_001191331) [132]. For h*PRPS2,* there are also two transcript variants 1 and 2 mRNAs (NM_001039091 and NM_002765), which differ only in length by nine nucleotides due to alternative splicing giving rise to two proteins, one of 321 aa and the other of 318 aa (RefSeq NP_001034180 and NP_002756) [43,45].

There are three classes of PRS enzymes, *viz*. I, II and III, with the hPRPS1 belonging to class I, which require inorganic phosphate (P_i_), SO_4_^2−^ and Mg^2+^ for enzymatic activity, are dependent on ATP or dATP as the diphosphoryl donor and are allosterically regulated by ADP, AMP and GDP [3,21,133]. Class II enzymes are specific to plants, and they accept a wide range of diphosphoryl donors (ATP, dATP, CTP, GTP, and UTP) and are not allosterically inhibited by ADP or dependent on P_i_ for their activity [134,135]. Class III enzymes are a novel class of PRPP synthetase found in the archaeon *Methanocaldococcus jannaschii,* which utilize ATP and dATP as a diphosphoryl donor and whose activation is solely dependent on P_i_. They are inhibited by ADP in a non-allosteric manner but appear to be competitively inhibited by ATP [136]. The PRPP synthetase from the thermophilic archaeon *Thermoplasma volcanicum* is an interesting case since on the basis of structure, it has a close similarity to *M. jannaschii* and therefore, has been designated as a class III PRPP synthetase, in spite of lacking an allosteric site and existing as a dimer [137]. In addition to the biochemical characterization of PRPP synthetases, they can also be classified according to their three-dimensional structures: class I enzymes have a hexameric quaternary structure found in mammals and bacteria; class II enzymes, including PRPP synthetase isozyme 4 from spinach, have a homo-trimer quaternary structure; and the class III enzyme from *M. jannaschii* is tetrameric [136]. The crystal structure of *B. subtilis* PRS (PDB: 1DKR) [21] has been known for twenty years and has provided the basis for structural analysis of hPRPS1 (PDB: 2H06) and is a homo-hexamer consisting of three tightly interacting homodimers [133]. Each of the homodimers contains a catalytic site and two allosteric sites, I and II, whereas, in the *B. subtilis* PRS, there is only one allosteric site [21,133]. The reactive site binds both ATP and ribose-5-phosphate and is located at the interface of two domains within one homodimer. The allosteric site I for ADP and P_i_ is positioned at the interface of the three homodimers of the hexameric structure, whereas the second, novel allosteric site II occupies a position at the interface of two monomers within one homodimer. The crystal structure of hPRPS1 reveals a complex with SO_4_^2−^ at the allosteric site II, thus positively impacting on the stability of the catalytic flexible loop and preventing the entry of the inhibitor ADP while maintaining an open conformation for the entry of ATP and activation of the enzyme [133,138].

Human PRPS1 and PRPS2 are class 1 PRPS enzymes that, despite their high degree of sequence similarity, differ in function and regulation. It has been observed that *PRPS1* is expressed constitutively, whereas *PRPS2* is expressed inter alia in colorectal cancer (CRC) metastasis and hepatocellular carcinoma (HCC) [139,140]. In addition to hPRPS1 and hPRPS2, human PRPS-associated proteins (PAPs) (PAP39 (hPRPSAP1, HGNC 9466) at position 17.q25.1 and PAP41 (hPRPSAP2, HGNC 9467) at position 17p11.2 have been discovered (Figure 8b) [13,17]. Human PAPs share high amino acid sequence similarity and interact physically with hPRPS1 [141,142,143,144,145]. However, PAPs, PRSPSAP1 and PRPSAP2 do not contain an ATP-catalytic binding site and, therefore, cannot be directly involved in the phosphoribosyl transferase reaction but may act as negative regulators of PRPP synthetase [13,17]. It should be noted that the insertions in rat *PRSPSAP1* and rat *PRPSAP2* are at similar positions to NHR1-1 of Sc*PRS1* and NHR5-2 of Sc*PRS5* (cf. Figure 2) [145]. Two processed pseudogenes for h*PRPS1* (HGNC 39427 and HGNC 9464) are located on chromosomes 2q24.3 and 9q33.3, respectively (Figure 8b), and both are transcribed but not translated. Human PRPS1-related diseases can be characterized by a continuous spectrum of features spanning neuropathies, hearing loss, optic atrophy, ataxia, cognitive impairment, and recurrent upper-respiratory infections [118]. It is possible to correlate mutations in h*PRPS1* with the severity of the disease, which ranges across varying degrees of loss of PRPS activity attributable to minor changes in the architecture of the hexameric structure. For instance, PRPS activity in the neuropathy CMTX5 patients is reduced in comparison to patients presenting with DFN2-associated hearing loss because of the nature of the local structural changes of the enzyme. There is also the possibility that epigenetic modification affects h*PRPS1* expression [146,147]. The neurological symptoms of h*PRPS*-related diseases may be associated with the degree of myelination of neuronal structures since the synthesis of myelin is dependent on lipid esters of pyrimidine nucleotides and CDP-choline and requires SAM (*S*-adenosylmethionine) [118]. We have observed that in the *prs1Δ* strain, the combined CDP and CTP levels were extremely low [26,76,80], only 12% of the level in the WT, thus connecting PRPP synthesis to both *de novo* and salvage synthesis of phospholipids which require CTP. The osmotic sensitive phenotype of a *prs1Δ* strain is rescued by overexpression of CTP synthetase [80,106]. Neurons have a high energy demand and are therefore susceptible to any reduction in GTP and ATP supply. There is a parallel scenario in yeast that CTP pools are important for maintaining the supply of CDP-choline and CDP-ethanolamine for phospholipid biosynthesis, essential for membrane integrity [118,148]—a further argument for the use of yeast as a eukaryotic model organism. The role of pyridine nucleotides in energy production cannot be ignored since they are involved in energy storage, cell signalling, nucleic acid structure and enzyme cofactors, e.g., NAD^+^ [118]. The conflicting evidence that PRPS1 superactivity can lead to purine depletion and overproduction of uric acid reflects the spectrum of mutations encountered in the product of the h*PRPS1* gene, which can affect local enzyme structure or impinge on the integrity of the quaternary structure and the stability and overall activity of the enzyme. 

### 4.1. PRPP Synthetase Superactivity: hPRPS1 (*OMIM 300661*)

Missense mutations of h*PRPS1*, the more highly expressed isoform of the two h*PRPS* genes, lead to gain-of-function mutants [46,118,149]. These changes cause critical alterations in PRPS levels that may have a significant influence on many life processes, such as nucleotide biosynthesis and cell signalling. The gain-of-function of h*PRPS1* results in superactivity of the enzyme associated with hyperuricaemia and hyperurocosuria—overproduction and accumulation of uric acid in the blood and urine, respectively. Overproduction of uric acid can lead to gouty arthritis in joints and impairment of kidney function because of excessive production of purine nucleotides and their subsequent conversion to uric acid [50] and references therein [119,121,125,150]. Specifically, PRPS1 superactivity is a rare X-linked disorder of purine metabolism that results in an increase in enzyme activity due to regulatory defects or alteration in catalytic properties [14,149,151,152,153,154,155]. Overexpression of h*PRPS1* with altered kinetic enzyme characteristics has been observed in white cells, erythrocytes and fibroblasts of patients [125,154,155] and is characterized by several physiological disorders, including, in addition to gouty arthritis, hearing impairment or deafness, hyperuricaemia, hyperuricosuria, urolithiasis in men, ataxia, hypotonia, developmental delay, intellectual impairment and recurrence of infectious diseases in the upper respiratory tract [118,119,125,156,157,158,159]. Since the h*PRPS1* gene is located on the X-chromosome, if one of the mother’s two X-chromosomes carries the mutation for PRPS superactivity and this chromosome is inherited by the son, he will suffer this disease, and he can pass the affected chromosome to a daughter. The daughter will not necessarily display symptoms since she may have inherited an unaffected chromosome from her mother. Women are less likely to suffer from PRPS1 superactivity since random X-chromosome inactivation [160,161] may reduce the dosage or eliminate the mutated version of the h*PRPS1* gene.

Patients with severe or extremely high PRPS1 superactivity may exhibit one or more of the following nine missense *PRPS1* mutations, p.D52H (c.154G > C), p.L129I (c.385C > A), p.A190V (c.569C > T), p.H193Q (c.579C > G) [151], p.N114S (c.341A > G) and p.D183H (c.547G > C) [118,151], p.H193L (c.578A > T) [150,151], p.G174V (c.521G > T) [162] and p. G174R (c.520G > A) [163]. Specifically, the missense mutations of p.D52H and p.L129I result in the disruption of local structure close to the allosteric sites and therefore inhibit feedback regulation [118,151,164]. The p.D52H mutation causes destabilization of the enzyme structure around Asp52 and directly disturbs allosteric site I [118,120,147], thus reducing the sensitivity to the inhibitor ATP in vivo [164], whereas the p.L129I mutation leads to steric hindrance with both the protein backbone at Ala131 and Ile134 disturbing allosteric site II [118], leading to a neuropathy [147,151]. 

The other missense mutations are located in the homodimer interface, thereby disrupting the homodimer and affecting the allosteric sites [119,150]. The p.N114S mutation also destabilizes the hydrogen bonding of the homodimer. The p.A190V mutation leads to disruption of the hydrophobic environment in the homodimer interface, whereas p.D183H and p.H193L/Q mutations break the hydrogen bond interactions between Asp183 of one homodimer partner and His193 of the other polypeptide chain of the homodimer, thereby destabilizing the homodimer interface [118,147]. Interestingly, the h*PRPS1* mutations p.D183H and p.N114S in both hemizygous males and affected females caused neurological deficiency, e.g., sensorineural deafness [50,120], whereas p.D52H leads to increased production of PRPP levels and induces hyperuricaemia and gout [164]. In males, PRPS1 superactivity resulted in an alteration in the allosteric control of PRPP synthesis with high purine nucleotide and uric acid production [17]. However, the study of Zikánová et al. (2018) [163] showed that female carriers of PRPS1 superactivity had elevated levels of uric acid and gout since random X-chromosome inactivation may result in them retaining only the mutated allele p.G174R (c.520G > C).

### 4.2. Reduced PRPP Synthetase in Humans

Reduced activity in *PRPS1* is associated with neurological disorders, such as Arts syndrome, Charcot-Marie-Tooth disease type 5 (CMTX5), X-linked syndromic/nonsyndromic sensorineural deafness (DFN2/DFNX2) and retinal dystrophy.

#### 4.2.1. Arts Syndrome

This disorder (OMIM 301835) is associated with X-linked missense mutations in h*PRPS1*, resulting in loss-of-function of PRPS1. It affects mainly males with a prevalence of less than 1 in 10^6^ and is lethal, with symptoms appearing before two years of age. Unfortunately, 80% of patients die in childhood. It is a severe phenotype caused by the missense mutation in c.455T > C (p.L152P), which was first identified in a Dutch family. A further missense mutation c.398A > C (p.Q133P) was identified in an Australian family [47,48]. MD revealed that the p.Q113P variant destabilizes the allosteric site II and the ATP binding pocket of hPRPS1 and that p.L152P alters only the ATP binding site [48]. An autopsy of the Dutch Arts syndrome patient revealed a total lack of myelin in the posterior columns of the spinal cord, and a sural nerve biopsy of the Australian patient showed mild paranodal demyelination. Arts syndrome is characterized by profound delayed motor development, early-onset hypotonia, hearing impairment, mental retardation, optic atrophy, intellectual disability, a compromised immune system, recurrent infection, and early childhood mortality [47,48,118].

Maruyama et al. (2016) [165] also reported a transient neurogenic proximal weakness and loss of muscle strength caused by the mutation p.H123D (c.367C > G) in Arts syndrome patients. The premature death of Arts syndrome patients is most commonly due to infections of the upper respiratory tract [47]. In this disorder, low serum uric acid concentrations and undetectable hypoxanthine with normal range xanthine and uric acid concentrations in the urine purine profile have been observed [48]. In affected males, PRPS enzyme activity is significantly decreased, while obligate female carriers show mild symptoms [166]. PRPP synthetase activity, as measured in fibroblasts from the affected members of the Dutch family, was reduced 10-fold in comparison to the activity from controls [47]. In a recent study, Puusepp et al. (2020) [167] reported no hearing loss and normal levels of both serum uric acid, purines and pyrimidines in a hemizygous variant c.130A > G (p.I44V) of the h*PRPS1* gene in a 6-year-old male patient presenting with clinical features of Arts syndrome, as described above. However, a marked reduction in PRPS1 activity in erythrocytes was recorded in this patient, emphasizing the necessity of testing for all parameters associated with Arts syndrome. It is noteworthy that the p.I44 residue is conserved in mammals and amphibia, *Drosophila* spp., *Caenorhabditis elegans* and *S. cerevisiae*.

Another mutation, c.424G > C (p.V142L), was found in a patient presenting with classical symptoms, including hyperuricaemia and hyperuricosuria but with no gouty arthritis, developmental delay, hypotonia, and bilateral hearing loss, indicative of both the severe form of hPRPS1 superactivity and symptoms of Arts syndrome, e.g., recurrent respiratory infections, myopia, glaucoma and motor neuropathy [157]. This valine-to-leucine change arising from a transversion is predicted to alter the allosteric sites I and II and the ATP binding site, thus impairing hPRPS1 inhibition and resulting in hPRPS1 overexpression. Furthermore, this substitution is postulated to influence the interaction of Val142 with Lys100 in a flexible loop, which is part of the ATP-binding site of the other PRPS1 subunit of the homodimer. The two-fold effect of this mutation on nucleotide concentrations is reflected in the difference in PRPS activity measured in fibroblasts and postmitotic cells, e.g., erythrocytes and brain cells. In the former, there is increased PRPS1 activity due to loss of feedback inhibition, whereas in the latter, there is loss of PRPS1 activity over time. This combination of gain-of-function and loss-of-function of PRPS1 activity in the same patient was revealed by molecular modelling and enzyme activity measurement in different cell types and illustrated the range of features associated with an imbalance of PRPP synthetase underlying the clinical features of various non-syndromic postlingual hearing loss neuropathies. 

In summary, there are several manifestations, e.g., PRPS1 superactivity and recurrent infections, which can be traced back to the p.V142L mutation resulting in intermediate phenotypes lacking the severity of those associated with Arts syndrome (p.Q133P or p.L152P). There was a difference in PRPP synthetase activity as measured in erythrocytes and fibroblasts in the proband p.V142L, which could be due to the altered regulatory properties of the enzyme. Supporting evidence for this suggestion is that the GDP/GTP levels in the proband’s erythrocytes were reduced in comparison to the controls, but ATP/ADP levels were normal. Unfortunately, there was no data concerning GTP/GDP levels in fibroblasts, although enzyme activity in fibroblasts carrying the mutation did respond to the level of phosphate, whereas the control did not. It could be that intracellular purine levels are sensed by an as-yet-unknown mechanism(s) involving GTP interaction with mTORC1 *via* Rheb [168]. 

Overlapping phenotypes are also described in a male/female sibling study [147]. A male subject, aged 36, p.Q277P (c.830A > C), displayed overlapping features of CMTX5 and Arts syndrome, prelingual hearing loss, recurrent severe infections and progressive visual loss due to optic atrophy, as well as undetectable PRPS1 activity in erythrocytes. His sister had reduced PRPS1 activity in addition to prelingual nonsyndromic hearing loss and deafness (DFN2/DFNX1), whereas their mother had normal PRPS1 activity and no hearing loss. Brain MRI (magenetic resonance imaging) of the two siblings revealed cerebellar atrophy. X-chromosome inactivation was extremely skewed in the sister but only moderately skewed in the mother. Recently, in 2021 [169], a novel Arts syndrome h*PRPS1* deficiency missense mutant p.R84T (c.250C > T) was reported, which disturbed the interface regions between the subunits, thus negatively impacting the ligand binding of the allosteric site I. An Australian male patient presented in his first year with symptoms congruent with Arts syndrome, including congenital sensorineural deafness, recurrent severe infections, visual impairment inter alia with substantially reduced hPRPS1 activity in erythrocytes. It was decided with parental consent to start feeding studies with SAM (*S*-adenosylmethionine) and NR (nicotinamide riboside) at the age of three. Unlike most purines, SAM can cross both the gut and the blood-brain barrier and has been used successfully in the treatment of depression and dementia [170]. It was also shown that SAM, which can supply ATP and GTP independently of PRPP [118], may alleviate some of the symptoms of patients with Arts syndrome. NR can form NADP(H) independently of PRPP by virtue of NR kinases 1 and 2 to produce NMN (nicotinamide mononucleotide), which is adenylated to NAD^+^ [171,172]. Shortly after commencing treatment, the intervals between respiratory infections increased and recovery was improved. The patient’s muscular strength, speech, ability to play and overall well-being improved markedly. No adverse treatment effects were recorded, and homocysteine/methionine concentrations were normal. Despite these improvements, audio-visual impairment and ataxia persisted, but nevertheless, the treatment reduced the strain on the patient and his family. ATP and GTP improved sustainably, but NAD^+^ measurements could not be repeated with higher levels of NR [118,169]. 

#### 4.2.2. Charcot-Marie-Tooth Inherited Neuropathy (CMTX5)

CMTX5 (OMIM 311070) is a clinical and heterogeneous inherited neurological disorder that is caused by loss-of-function mutations in the h*PRPS1* gene or decreased activities of the hPRPS1 protein [173]. The missense mutations of h*PRPS1* occur at c.129A > C (p.E43D) in patients with Rosenberg-Chutorian syndrome [174], and the CMTX5 mutation p.M115T occurred in a Korean patient [51]. Both mutations influence the ATP binding pocket, but only p.M115T was predicted to potentially interact with the allosteric site I [51,118,147]. CMTX5 is one of the most common causes of inherited neuropathy in humans, and it has been estimated that approximately 1 carrier is found among 2500 people [147,175]. It is characterized by peripheral neuropathy, loss of muscle tissue and touch sensation, chronic motor and sensory impairments, hearing loss, optic atrophy and deafness [51]. CMTX5 is not associated with mental retardation and recurrent infections. This wide spectrum of phenotypes arising from mutations in the h*PRPS1* gene causes complications in attributing the phenotypes to specific mutations. The expression of the uric acid levels of the carriers is within the accepted normal range. A novel missense variant with an amino acid change p.M115V (c.343A > G) was identified in a 12-year-old Italian male who had post-lingual bilateral hearing impairment, which worsened upon ageing; PRPS1 enzyme activity in erythrocytes was reduced to less than 10% of the control. Interestingly, the female carriers lack or exhibit only mild symptoms of CMTX5, and no evidence of reduced life expectancy has been reported among the female carriers [51,176]. A further mutation p.V309F caused by the transversion c.925G > T located in exon 7 of the h*PRPS1* gene was found in a Peruvian family, in which the females were found to have subclinical signs of peripheral neuropathy, whereas the males displayed more evidence of mild symptomatic peripheral neuropathy. Furthermore, the PRPS1 activity in erythrocytes was drastically reduced to approximately 10% of the activity measured in an unaffected male [176]. The M115 residue may therefore be critical for enzyme activity since it has been noted in a previously identified CMTX5 family. The mutation p.M115T, which is in the N-terminal α-helical domain of the enzyme, may therefore destabilize the ATP binding site and the allosteric site I. Indeed, enzyme activity measurements in patient fibroblasts demonstrated a reduction of over 50% of PRPS activity measured in unrelated controls [51].

Almoguera et al. (2014) [146] stated that optic atrophy and *retinitis pigmentosa* were the major characteristics of the affected female carriers observed in a Spanish family. However, at the age of 10–12 years, the onset of peripheral neuropathy manifests with foot drop or gait disturbance as the initial symptoms. On the other hand, at the age of 7–20 years, the onset of bilateral progressive visual impairment presents with optic disc pallor and reduced visual potential among patients [50]. In contrast, Park et al. (2013) [177] reported that a male proband of a Korean family diagnosed with X-linked recessive CMTX5 did not reveal optic atrophy, cognitive impairment, visual disturbance, or respiratory dysfunction. By next-generation sequencing (NGS) analysis, Lerat et al. (2019) [178] established a mutation c.202A > T (p.M68L) in a 35-year-old male presenting with hearing impairment and CMTX5, both related to bilateral optic neuropathy but without any manifestation of *retinitis pigmentosa*. The first report of CMTX5 in a Chinese population has been published [179], describing that the c.344 > C (p.V112L) mutation in a thirteen-year-old boy was associated with congenital non-syndromic sensorineural deafness. At the age of three, this boy had developed progressive distal weakness of all limbs with muscle atrophy of hands and thighs, and virtually no PRPS1 synthetase activity could be measured in erythrocytes. A further hemizygous mutation c.82G > C (p.G28R) has been reported in a Japanese male with CMTX5 with typical clinical features, i.e., neuropathy, optic atrophy, and hearing loss, but with only mild loss of PRPS1 activity in erythrocytes [180].

#### 4.2.3. Sensorineural Deafness 2 (DFN2/DFNX1)

DFN2/DFNX1 (OMIM 304500) is a phenotypically complex disorder with p.D65N (c.193G > A), p.A82P (c.244G > C), p.A87T (c.259G > A), p.I275T (c.824T > C) p.I290T (c.869T > C), p.G306R (c.916G > C) causing nonsyndromic X-linked sensorineural deafness [49,50,118,147,156,181,182,183]. This disorder accounts for approximately 5% of congenital deafness and less than 2% of nonsyndromic hearing impairment [49]. Mercati et al. (2020) [184] identified the following missense mutations, p.R196W (c.586C > T), p.D128V (c.383A > T), p.E69A (c.206A > C) and p.E46Q (c.136G > C). The study consisted of individuals from four unrelated families which are all associated with profound deafness. All four probands were diagnosed with bilateral sensorineural hearing loss. Probands carrying p.D128V and p.E69A presented additionally with CMTX5 and p.R196W with retinal dystrophy (see Section 5., Retinal Dystrophy). Further mutations p.E43D and p.M115T have been shown to be responsible for combination of hearing impairment and mild peripheral neuropathy [182]. Interestingly, there is also a link to gain/loss-of-function in the following mutations: p.N114S, p.D183H, p.Q133P, p.L152P and p.V142L which all are associated with profound deafness. The first two-named also cause gain-of-function, i.e., PRPS1 superactivity whereas p.Q133P and p.L152P give rise to both Arts syndrome and congenital hearing loss. The mutation p.V142L causes an intermediate phenotytpe of PRPS1 superactivity and Arts syndrome [50,157]. From the data presented, it is clear, that there is a plethora of phenotypes associated with missense mutations in PRPS1 which is likely to increase in the future with technological advance and improved diagnostic approaches.

The mutations p.D65N, p.A87T, p.G306R and p.I290T may result in any of the following disorders: prelingual or postlingual, mild or moderate, bilateral, and progressive phenotypic hearing impairment [183]. However, none of these mutations cause a profound change in the enzyme structure, but through molecular modelling, it has been observed that a mutation of p.D65N affects the ATP binding site [49,118]. With the help of in vitro enzyme assays, a 45–56% reduction in PRPS activity has been reported in erythrocytes and cultured fibroblasts of DFN2/DFNX1 patients when compared to their respective normal controls [49,50]. Kim et al. (2016) [182] reported that the h*PRPS1* DFNX1 c.244G > C (p.A82P) mutation resulted in decreased PRPS1 enzyme activity in erythrocytes and audiologic phenotypes. They also observed that DFNX1 accounts for 2.4% of moderate sporadic sensorineural hearing loss (SNHL) in a Korean paediatric population. It is important to note that the mechanism of hearing impairment in DFN2/DFNX1 is yet to be elucidated. A further missense mutation, uncovered in an Italian family in which serine is substituted for alanine p.A113S (c.337G > T) is responsible for X-linked hearing loss, but with no symptoms of neuropathy. The mutation, which segregated with postlingual, bilateral progressive deafness, was identified in two brothers and their mother and caused a 70% reduction in PRPP synthetase activity as measured in the erythrocytes of the two brothers. It has been suggested that the mutation may destabilize the α-helix participating in the trimer interface and reduce ATP binding, as revealed by molecular modelling [176]. Screening of the h*PRPS1* gene in sixteen unrelated subjects from X-linked deaf families uncovered two additional missense mutations giving rise to aa changes p.M115V and p.V309F segregating with hearing impairment and associated with mild peripheral neuropathy. These mutations also resulted in a reduction of >60% PRPS1 activity as measured in erythrocytes. Mutations p.M115V and p.V309F are located in the trimeric interface of PRPS1 and destabilize the ATP-binding site and the allosteric site I. The mutation p.A121G (c.362C > G), which has been associated with CMTX5 and early-onset hearing loss but not with optic atrophy, may reflect a loosening of the interaction between the hexamers or a minimally ordered catalytic site [177]. Further identification of diseases correlating with h*PRPS1* mutation will be useful for the molecular diagnosis of clinically heterogenous peripheral neuropathies. However, the high sequence similarity in eukaryotic *PRPS1* genes offers the opportunity of studying inborn errors of metabolism in a variety of model eukaryotic organisms, e.g., *S. cerevisiae* and *D. rerio*.

## 5. Retinal Dystrophy

hPRPS1 synthetase variants have also been shown to be responsible for inherited retinal dystrophy (IRD). IRD has a heterogenous aetiology with over 300 disease-causing genes identified (The Retinal Information Network) and affects about 3000 individuals. It is the most common cause of blindness in the working age population in the UK. In an extensive study of nine affected females from five unrelated families, in which none of the males were affected, NGS revealed the following missense variants in the h*PRPS1* gene: c.46A > T + c.47G > T (p.S16F), p.R196W, c.641G > C (p.R214P), c.640C > T (p.R214W) [185]. These variants were detected in females presenting with IRD with interocular asymmetry, suggesting that alterations in h*PRPS1* may be the cause of IRD. It should be noted that proband p.R196W, described in Section 4.2.3., also presented with severe developmental delay, lack of speech development and *Diabetes insipidus* at the age of five [184]. A similar spectrum of *PRPS1*-related disorders has been described in Al-Maawali et al. (2015) [186].

Multiple sequence alignment confirmed that the serine residue at position 16 and the two arginine residues at positions 196 and 214 are highly conserved across different species. The arginine residue at p.R214 is also conserved in Prs1-Prs5 of *S. cerevisiae*. It is worth noting that mutations in p.S16P and p.R214W have also been associated with hearing loss, emphasizing the importance of PRPS1 in overlapping human diseases, i.e., syndromic hearing loss and Arts syndrome (p.Q133P) [48,50]. Each of the IRD variants is predicted to disrupt the stability or function of PRPS1. Normally, female carriers of X-linked recessive mutations are asymptomatic, but in the study by Fiorentino et al. (2017) [185], the retinal degeneration observed in females came to light on account of tissue-specific skewed X-chromosome inactivation or reduced levels of PRPS1 activity. The absence of affected males in the family studied could be explained by male embryonic lethality when male embryos inherit the X-chromosome carrying the mutations studied. A further mutation of codon 16 in exon 1, p.S16P (c.46A > C + c.47G > C) of PRPS1, when mutated at either the first or second position in the codon, gives rise to proline or phenylalanine p.S16F, respectively, which causes hPRPS1 deficiency in females [146]. The p.S16P mutation is in the first α-helix of the protein of the N-terminal domain, and it is known that the replacement of serine by proline is associated with helix breakage and, in both instances, causes retinopathy and hearing loss. 

More recently, the first evidence of sector *retinitis pigmentosa,* which is known to have a more favourable prognosis compared to generalized *retinitis pigmentosa,* has been linked to the mutation p.S16F in PRPS1 [187]. By means of NGS, a single variant in h*PRPS1* c.292G > A (p.D98N) was identified in a study of 100 nonsyndromic and syndromic patients with a clinical diagnosis of retina-linked blindness [188]. Taking into account ethical considerations, this opens the opportunity for CRISPR/Cas9-associated gene therapy for specific pathogenic variants of individual genes. Interestingly, the corresponding residue S_16_ in murine PRPS1 and PRPS2 is a target of the ataxia-telangiectasia-mutated kinase (ATM) in response to ionizing radiation. Notably, the phosphorylation of residue T_228_ in mPRPS1 and mPRPS2 is dependent on S_16_ phosphorylation to preserve the high activity of mPRPS1 and mPRPS2, thus providing further evidence for the role of PRPP in the innate immune response [189].

## 6. Roles of PRPS in Cancer

A recent development is the discovery of the involvement of PRPP synthetases 1 and 2—hPRPS1 & hPRPS2—in cancer progression, management, and treatment. Human PRPS1 and PRPS2 have been identified in acute lymphoblastic leukaemia (ALL) as the major cause of drug resistance to the prodrugs thioguanine and thiopurine, which are used in cancer therapy [189,190,191,192]. The way in which cancer cells become resistant to the prodrugs is through the acquisition of hyper-activating h*PRPS1* mutations, e.g., c.568G > A (p.A190T), p.A190V, c.340A > G (N114D), p.D183H, c.521G > A (p.G174E), c.528A > T/C (p.K176N), c.571C > T + c.573G > T/C (p.L191F), c.308G > C (p.S103T) ([193] and other mutations cited therein). All these mutations destabilize the dimer interface, causing the loss of ADP and GDP allosteric inhibition. Large-scale sequencing of DNA from patients with therapy-resistant mutations in relapsed ALL showed the presence of h*PRPS2* hyper-activating mutations c.400G > A (p.A134T) and c.523G > A (p.A175T) [190]. h*PRPS1* and h*PRPS2* have been implicated in the metastasis and proliferation of cancer [193,194]. Mammalian PRPS2 is associated with oncogene Myc-driven tumorigenesis activation [52] and promotes metastasis of neuroblastoma cancer [194]. The translationally regulated enzyme PRPS2 may play a role in the proliferation of oncogenic signalling-driven cancers. This is due to an eIF-4E cis-regulatory pyrimidine-rich translational element (PRTE) found in m*PRPS2* and h*PRPS2* [52,195], but not in the corresponding *PRPS1* genes, which renders PRSP2 resistant to feedback inhibition by ADP and GDP and may also explain the therapy-resistant human cell lines associated with ALL [190]. The distinguishing features of the *PRPS2* isoform are found in many translationally regulated transcripts because of mTOR hyperactivation, allowing nucleotide and protein biosynthesis to increase concomitantly with higher protein synthetic capacity protein of cancer cells [52,196,197]. *PRPS2*-knockout mice are viable and fertile with no discernible developmental defects, which supports the hypothesis that mPRPS2 is not essential for development but contributes to Myc-dependent cancer formation. However, a recent publication [198] has correlated the depletion of murine *PRPS2* by sh RNA lentivirus with hypospermatogenesis, suggesting that PRPS2 may be a potential biomarker for male infertility. Human *PRPS2* may well be an attractive diagnostic target for Myc-driven cancers in osteosarcoma [199]. The c-Myc-PRPS2 interaction promotes metabolic reprogramming of biosynthetic processes leading to the proliferation of cancer cells. There is statistical evidence linking a longer survival time with negative PRPS2/p-mTOR staining in comparison to positive PRPS2/p-mTOR staining, implying that Myc-driven PRPS2 expression *via* the eukaryotic translation initiation factor 4E (eIF-4E) allowing interaction with mTOR results in tumour progression.

Overexpression of hPRPS1 elevates cancer cell proliferation and inhibits apoptosis in B-cell acute lymphoblastic leukaemia cell lines (B-ALL) as well as increasing the expression of Bcl-2, a gene in the mitochondrial-associated apoptosis pathway, at the RNA and protein levels [200]. The same authors suggested that h*PRPS1* silencing has no effect on the regulators of the cell cycle but elevates apoptosis in B-ALL cell lines. He et al. (2017) [201] found that h*PRPS1* silencing increases apoptosis in breast cancer. The expression of h*PRPS1* and h*PRPS2* is reduced by hypoxia in glioma cells. Furthermore, the expression of *PRPS1*, *PRPS2*, *PRPSAP1* (*PAP39*) and *PRPSAP2* (*PAP41)* genes in U87 glioma cells is not inhibited by ERN1 (EC 2.7.11, endoribonuclease activity of endoplasmic reticulum to nuclei-1). Endoplasmic reticulum-induced stress *via* tunicamycin in glioma cells, together with the suppression of ERN1, downregulates *PRPS1* and *PRPS2* gene expression levels [202]. The combined deletion of *PRPS1* and *PRPS2*, *PPAT* (PRPP-amidotransferase, EC 2.4.2.14) (HGNC 9238) and *CAD* (carbamoylphosphate synthetase II, EC 6.3.5.5) (HGNC 1424) suppresses cell proliferation in Kaposi’s sarcoma-associated herpes virus (KSHV), by redirecting glutamine and asparagine, which provide the nitrogen source for purine and pyrimidine biosynthesis from the TCA cycle to the biosynthesis of nucleotides and non-essential amino acids, suggesting potential therapeutic application for this pathway [203]. 

Interestingly, in a recent study conducted by Miao and Wang (2019) [139], it has been reported that hPRPS2 promotes the movement and invasion of colorectal cancer cells (CRC), suggesting that the upregulation of PRPS2 observed in CRC was induced by the *MYC* proto-oncogene, thus implicating PRPS2 in CRC metastasis. Previously, Qiu et al. (2015) [54] demonstrated that the mRNA and protein levels of h*PRPS1* are upregulated in CRC. Knockdown of h*PRPS1* or upregulation of the micro-RNA, miR-124, a tumour suppressor, leads to reduced DNA synthesis and cell proliferation in CRC cell lines, revealing potential therapeutic targets. There are miRNA integrated signatures for h*PRPS1* mRNA and h*PRPS2* mRNA interactions, which may represent potential biomarkers and therapeutic targets for CRC [204].

MD simulation of four missense mutations p.D52H, p.M115T, p.L152P and c.607G > C (p.D203H) in h*PRPS1* predicted to be disease-causing mutations [175], revealed that the CMTX5 mutation p.M115T affected protein stability and is consistent with previously published experimental results documenting 60% reduction in hPRPP synthetase activity due to the disruption of the ATP binding region and allosteric site I of the PRPS1 active site [51]. Interestingly, a further h*PRPS1* missense mutation c.335T > C (p.V112A) spans the reference SNP (rs11541075) for the BOR (Branchio-Oto-Renal) syndrome. This syndrome, which is associated with hearing loss and other hearing abnormalities, in addition to abnormalities in kidney structure and function, may affect approximately 1 in 40,000 people (MedlinePlus). These data show the far-reaching effects of missense mutations in the h*PRPS1* gene which, on reflection, are not unexpected in a gene whose product is required to produce essential metabolites involved in nucleotide synthesis as well as contributing to cellular signalling and energy metabolism [175,205].

KHK-A (ketohexokinase-A), a protein kinase that is a splicing variant of KHK-C, an enzyme involved in fructose metabolism, phosphorylates and activates hPRPS1 at T_225_
*via* the inhibition of ADP, AMP, and GDP by sterical blocking allosteric site I to ADP, thus leading to an increase in nucleotide synthesis and progression of HCC. The oncogenic transcription factor c-Myc upregulates hnRNPH1/2 expression, thereby favouring splicing from KHK-C to KHK-A which then phosphorylates PRPS1 at T_225_, increasing PRPP synthesis. By in vitro phosphorylation assays it was shown that KHK-A functions as a protein kinase rather than phosphorylating fructose, thus constitutively activating PRPP production in HCC which has been linked to a considerable risk of poor prognosis in HCC [206]. Clinical relevance has been demonstrated by immuno-histochemical comparative studies of matched tumour and non-tumour liver tissue samples supporting the molecular biological studies [206]. The upregulation of PRPS1 by KHK-A has also been shown to increase the growth of oesophageal squamous cell carcinoma [207]. Furthermore, the suppressed tumour formation of glioma CD133^+^ cells by reduction in PRPS1 expression is a result of increased cell apoptosis [208]. Screening of a miRNA-targeting database identified *PRPS1* mRNA as being negatively regulated by the expression of miR-154 in glioma CD133^+^ cells [209]. This result has been echoed in neuroblastoma cells [210], adding weight to the potential of PRPS1 targeted therapy in glioblastoma multiforme and neuroblastoma cell proliferation. 

Jing et al. (2019) [211] observed a delay in the cell cycle and a reduction in CRC proliferation due to the lack of PRPS1 phosphorylation site at S_103_. CDK1 (cyclin-dependent kinase 1)-driven cell proliferation is dependent on the phosphorylation of PRPS1 at S_103_, possibly by blocking the binding of ADP to the allosteric sites of PRPS1. Upregulation of PRPS1 synthetase activity in the S-phase allows CRC cells to overcome cell cycle progression. Manipulation of this residue may be a promising strategy to combat CRC, especially since the mutations p.S103N (c.308G > A) and p.S103T have been shown to affect PRPS1 synthetase activity in ALL [193]. PRPS1 synthetase is regulated by the ratio of ATP/ADP which inter alia, indicates the metabolic capacity of a cell. ADP is the key allosteric inhibitor for PRPS1 in the presence of the substrate ribose-5-phosphate [133]. However, at a low concentration of ribose-5-phosphate ADP is a competitive inhibitor with respect to ATP, resulting in reduced PRPS1 activity and slowing down the progression of CRC. Nevertheless, further investigation is required prior to exploiting p.S_103_ for therapeutic purposes. 

Notably, PRPS2 can be arginylated on Asn3 (asparagine), distinguishing it from PRPS1 by conferring differential degradation for the two proteins. Arginylation affects both the activity and stability of PRPS2 since different molecular mechanisms are responsible for its degradation [212]. The discovery that the p300 KIX-(kinase-inducible) domain interacted with PRPS1 indicates that, in addition to its interplay with cAMP regulation, p300 may play a co-activator role in the regulation of DNA synthesis [213]. 

Qiao et al. (2020) [214] postulated that the knockdown of h*PRPS2* blocks the progress of prostate cancer *via* the suppression of the cell cycle and induction of apoptosis. They revealed that PRPS2 upregulates prostate adenocarcinoma tissues, whereas knockdown of h*PRPS2* suppresses cell growth in both in vitro and in vivo studies. Li et al. (2019) [210] reported that the down-regulation of h*PRPS1* in vitro and in vivo blocks neuroblastoma cell proliferation and tumour growth. Qian et al. (2018) [215] reported that glucose deprivation causes AMPK-mediated direct phosphorylation of PRPS1 at S_180_ and PRPS2 at S_183,_ which leads to brain tumorigenesis. Interestingly, under these conditions, the collapse of the hexameric structures of PRPS1 and PRPS2 to the constituent monomeric states occurs, along with concomitant loss of PRPS1 and PRPS2 synthetase activities, nucleic acid synthesis and NAD^+^ synthesis. 

The study conducted by Lei et al. (2015) [216] demonstrated that the expression of PRPS2 blocks the apoptosis of Sertoli cells, which correlates with Sertoli cell-only syndrome (SCOS). PRPS1 was found to be active in brain tumour-initiating cells (BTICs), indicating that PRPS1 is vital for maintaining BTIC linking MYC with *de novo* purine synthesis [217]. Wang et al. (2018) [218] reported that h*PRPS1* mutants elevate intracellular PRPP levels, which potentially enhances the transformation of 5-fluorouracil to fluorodeoxyuridine monophosphate and fluordeoxyuridine triphosphate, which are known metabolites that can induce DNA damage and apoptosis. The study suggests that ALL patients with h*PRPS1* mutations are likely to benefit from 5-fluorouracil-based chemotherapy.

## 7. Role of PRPS1 in Ageing

Hutchinson–Gilford progeria syndrome (HGPS), also known as progeria, arises from a mutation in the gene encoding lamin A, which is a protein scaffold at the inner edge of the nuclear membrane. It is a very rare fatal disease that causes accelerating ageing in infancy [219,220]. Cell lines from HGPS patients and their healthy parents were studied by RNAseq and iTRAQ (high-resolution quantitative proteomics) techniques, and the patients’ cells were found to have reduced levels of aa and a high level of glycolysis, leading to a reduction in ribose-5-phosphate levels, thus compromising PPRS1 activity and affecting *de novo* purine synthesis. Supplementation of the cell culture media with SAM improved cell growth and reduced the proportion of senescent cells. These observations are consistent with premature ageing being associated with reduced purine metabolism and open a new therapeutic opportunity for the treatment of progeria [221].

## 8. Conclusions

PRS plays a vital role in the biosynthesis of purine and pyrimidine nucleotides, NAD^+^ and NADP^+^ and the aa histidine and tryptophan, all essential in many life processes. However, PRPP-synthesizing enzymes are not found in certain intracellular parasites. Mutations of PRS in different model organisms result in the disruption of cellular processes and metabolic and neurological disorders. In humans, it has been established that mutations in h*PRPS1* lead to a loss or gain-of-function. Gain-of-function in h*PRPS1* causes PRPS1 superactivity *via* the alteration of allosteric sites I and II, affecting either the trimer or the dimer interfaces, whereas the loss-of-function mutations in h*PRPS1* result in the following neurological disorders: Arts syndrome, CMTX5 and DFNX1/DFN2. This occurs by destabilizing the ATP binding sites and disturbing allosteric site I and the trimer interface in some instances. Furthermore, h*PRPS1* and h*PRPS2* have been implicated in cancer; for example, h*PRPS2* knockout causes c-Myc-driven tumourigenesis and promotes the metastasis of neuroblastoma cancer, while overexpression of h*PRPS1* elevates cancer cell proliferation and inhibits apoptosis in B-ALL cell lines. The two PRPS isoforms have distinct roles in normal physiology and disease; however, the molecular mechanisms responsible for this differentiation are not yet wholly understood, although the kinetic parameters of the two isoforms have been known for over 20 years [222]. PRPS2 is more responsive to the cellular concentration of ATP and loss of PRPS2 is synthetically lethal when MYC is overexpressed and, indeed, essential for Myc-driven cancer proliferation, but dispensable for normal physiology [52]. As described in [157] PRPS activity in erythrocytes and fibroblasts differs significantly, a situation which has the potential to be exploited for cancer therapy [139,140]. The signalling functions attributed to nucleotides, impinge on the homeostasis of cancer cell proliferation [223].

In *S. cerevisiae*, negative and positive genetic interactions established that one of these three functional entities, Prs1/Prs3, Prs2/Prs5 and Prs4/Prs5, must be present for survival. *S. cerevisiae* cannot survive with only a single Sc*PRS* gene, indicative of the importance of Sc*PRS* genes not only in one aspect of yeast cellular metabolism but also by exploiting genome duplication to allow the acquisition of insertions which link PRPP synthesis with the maintenance of CWI. The simultaneous deletion of Sc*PRS1*/Sc*PRS5* or Sc*PRS3*/Sc*PRS5* causes synthetic lethality, whereas the knockdown of Sc*PRS1*/Sc*PRS3* in the absence of Sc*PRS2* or Sc*PRS4* leads to a reduction in enzymatic activity and decreased capacity to synthesize PRPP as well as severe growth retardation in *S. cerevisiae*. Y2H results were confirmed by co-immunoprecipitation, revealing that Prs1 interacts with Slt2, the MAPK of CWI, indicating that PRS enzymes are involved in PRPP synthesis and also play an essential role in the maintenance of CWI in *S. cerevisiae*. The CWI pathway does not stand alone in yeast metabolism but has crosstalk to other MAPK signalling pathways, e.g., the HOG (high-osmolarity glycerol) pathway [224,225]. Pbs2, the MAPKK of the HOG pathway physically interacts with MKK1 of the CWI pathway as do Slt2 and Hog1, the MAPK of the HOG pathway [224] under conditions of cell wall stress to form a complex, emphasizing that cell signalling should not be regarded as a linear phenomenon but rather more as an interconnected network at multiple levels of cellular metabolism. 

Zebrafish have two Dr*PRPS1* paralogues, *prps1a* and *prps1b,* with a high sequence similarity to h*PPRS1*. It was observed that the deletion of Dr*prps1a* and Dr*prps1b* resulted in smaller otic vesicles, otoliths, and loss of inner ear hair cells, indicating that the absence of Dr*prps1a* or Dr*prps1b* in zebrafish results in sensorineural hearing impairment. In mice, m*PRPS1* and m*PRPS2* knockouts have been created. Murine *PRPS1* was identified in a high-throughput screen for skeletal phenotypes as one of the four X-linked genes responsible for hemizygous lethality [226]. Notably, scientific information on m*PRPS* mutations is scarce (MGI, Mouse Genomic Information). Nevertheless, the mouse model may be relevant since it has been shown that miRNA, specifically miR-376a-5p, represses the m*PRPS1* transcript at multiple binding sites within its 3′-UTR in a tissue-specific manner. These binding sites are not present in the 3′-UTR of m*PRPS2*. The expression of mouse and human miR-376 RNAs has been documented in both developing embryos and adult tissues, including the cochlea [227,228]. Specifically, it was shown that circKIF2A, also known as has_circ_0129276, could function as a sponge by removing miR-377-3p to enhance PRPS1 expression. However, the silencing of the circular RNA, circKIF2A, modulates the formation of the miR-377-3p/h*PRPS1* transcript complex, thereby reducing PRPS1 activity, which in turn reduces cell proliferation, metastasis, and glycolysis [229]. A similar scenario involving miRNA species/*h*PRPS1 transcripts and DLEU1—lnc RNA deleted in lymphocytic leukaemia 1 [230]—or lncRNA HAS2-AS1 (HAS2 (hyaluronan synthase 2) antisense RNA 1) [231], which are found to increase progression in various cancers, e.g., glioblastoma multiforme, suggests a potential therapeutic target for glioblastoma multiforme or CRC by destroying the interaction of miRNA/h*PRPS1* mRNA *via* ‘mopping up’ the miRNA with a concomitant reduction in hPRPS1 activity. Each of these models has the potential to provide useful information on the role of PRPP synthetase in neurodegenerative diseases and cancer. 

Overexpression of h*PRPS1* or h*PRPS2* in combination with sh/siRNA in selected cell lines may provide an ideal tool to screen and test for potential drug targets. In a human neuroblastoma cell line with h*PRPS1* knockouts, Li et al. (2019) [210] reported that overexpression of h*PRPS1* resulted in poor neuroblastoma patient prognosis and a low patient survival rate. However, down-regulation of h*PRPS1* inhibited neuroblastoma cell proliferation and tumour growth, thus resulting in good neuroblastoma patient prognosis and an improved patient survival rate. The use of the cell lines Sup-B15 and Raji indicated that overexpression of h*PRPS1* accelerates growth and inhibits apoptosis and may be responsible for an adverse/poor prognosis in Chinese children with B-ALL [52,192,200]. Investigation of established cell lines should be useful in identifying novel therapeutics against PRPS1 and exploiting the differential regulation of h*PRPS1* and h*PRPS2*. Each of the model organisms discussed has the potential to contribute useful information on the roles of PRPS1 and PRPS2 across the spectrum of eukaryotic PRPP synthetase.

In this respect, our collection of Sc*PRS* mutations demonstrating unexpected phenotypes and protein/protein interactions could prove useful in drug testing and uncover upstream regulators of PRPP in yeast, thus contributing to the role of PRPP in basic metabolism.

There is now compelling evidence that h*PRPS* mutations cause neurological disorders and resistance to cancer therapy in humans. It should be noted that, in spite of *D. rerio*’s ability to grow continuously, the zebrafish model has the potential to be used in studying nonsyndromic X-linked sensorineural deafness, retinopathies and neurological disorders associated with h*PRPS1* mutations. 

The elucidation of the signalling pathways in the model organisms described is essential for their exploitation. In the higher eukaryotes, the connection between genotype and phenotype is more complex than in *S. cerevisiae*. However, the connections uncovered in yeast (Figure 9) may inform investigations to be undertaken in zebrafish. 

## Figures and Tables

**Figure 1 cells-11-01909-f001:**
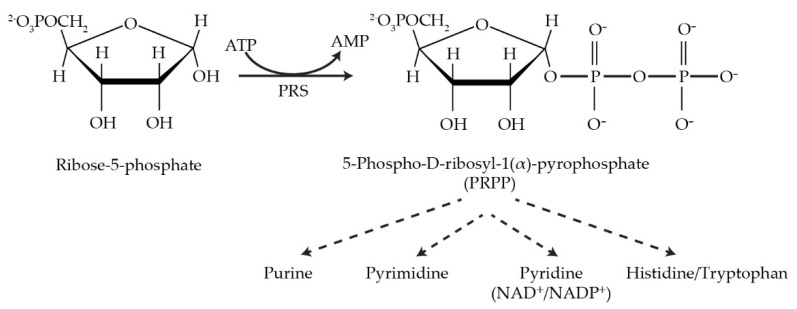
The reaction catalyzed by PRPP synthetase and metabolic pathways utilizing its product, PRPP.

**Figure 2 cells-11-01909-f002:**
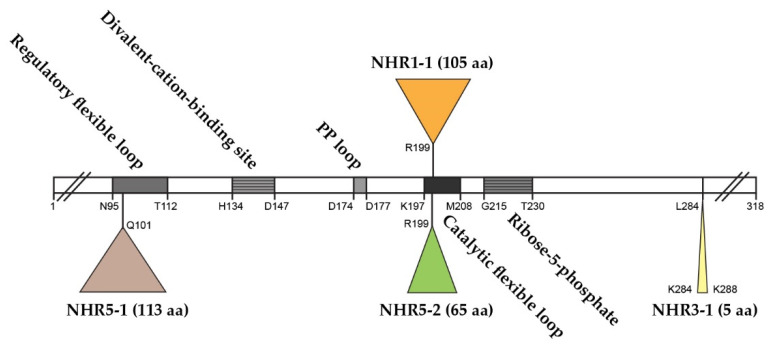
Relative positions of the non-homologous regions—NHR1-1, NHR3-1, NHR5-1 and NHR5-2—in the Prs2 polypeptide as the prototype. The Prs2 polypeptide consists of 318 aa and is indicated as an open bar with characteristic motifs therein. PP = pyrophosphate. The NHRs are positioned in Prs2 as triangles above or below the open bar. The coordinates of the four insertions reflect their lengths in Prs1, Prs3 and Prs5, respectively. The insertion points are defined where the similarity between Prs2 and Prs1 or Prs5 falls off. Sequence comparison was performed using the pairwise sequence alignment programme EMBOSS Needle.

**Figure 3 cells-11-01909-f003:**
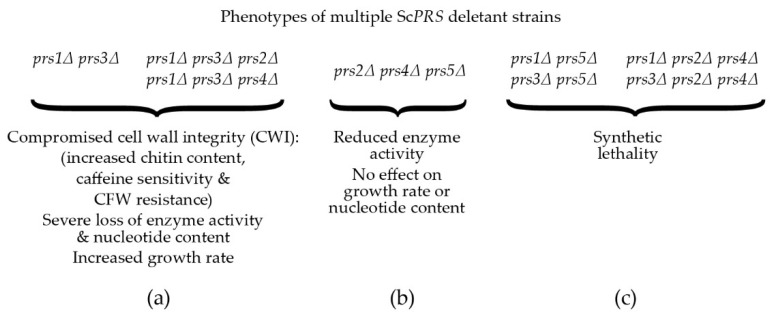
Compromised phenotypes, PRPP synthetase activity and nucleotide content encountered in Sc*PRS* multiple deletant strains: (**a**) phenotypes of defined Sc*PRS* deletant strains. CFW = calcofluor white; (**b**) phenotypes of the triple deletant *prs2Δ prs4Δ prs5Δ*; (**c**) synthetic lethality encountered when either Sc*PRS1* or Sc*PRS3* is deleted from a *prs5Δ* strain or simultaneous deletion of Sc*PRS2* and Sc*PRS4* in combination with loss of either Sc*PRS1* or Sc*PRS3*.

**Figure 4 cells-11-01909-f004:**
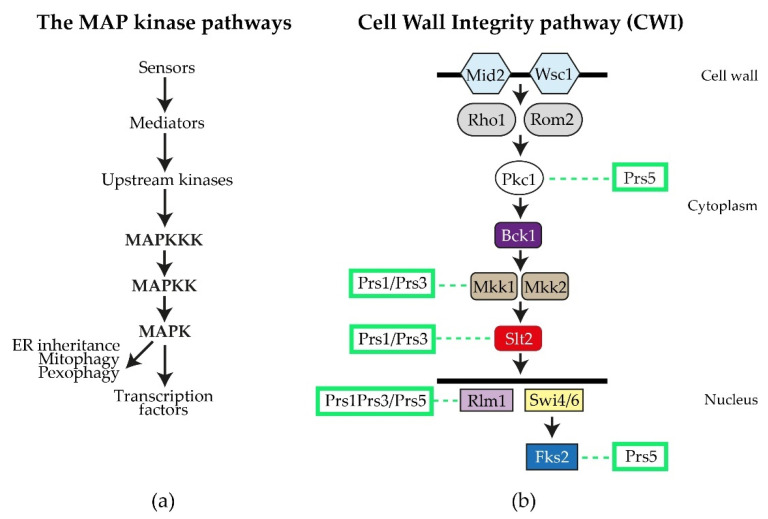
The *S. cerevisiae* CWI (cell wall integrity) pathway: (**a**) the general flow of information for MAP kinase pathways; (**b**) schematic diagram of the yeast CWI pathway. The two sensory proteins Mid2 and Wisc1 span the cell wall and are responsible for receiving signals, e.g., heat shock, that are transmitted to the upstream kinase Pkc1 by the mediators Rho1 and Rom2. Pkc1 then activates Bck1 (MAPKKK), which in turn activates Mkk1/Mkk2 (MAPKK), permitting the activation of Slt2 (MAPK). Slt2, in its phosphorylated state, can enter the nucleus, where it influences gene expression by targeting transcription factors, such as the MADS-box transcription factor Rlm1 and the SBF (Swi4/Swi6) transcriptional complex, as well as impacting on polarized growth and processes associated with mitophagy, pexophagy and ER inheritance. Fks2 is an alternative catalytic subunit of the β-1,3-D-glucan synthase whose expression is regulated by cell wall stress and Rho1 [83,84]. The interactions of Prs polypeptides with components of the CWI pathway or their impact thereon, are collated from the following references [64,74,78,79,80,84,92,95,96,97,98,99] and indicated by dashed lines.

**Figure 5 cells-11-01909-f005:**
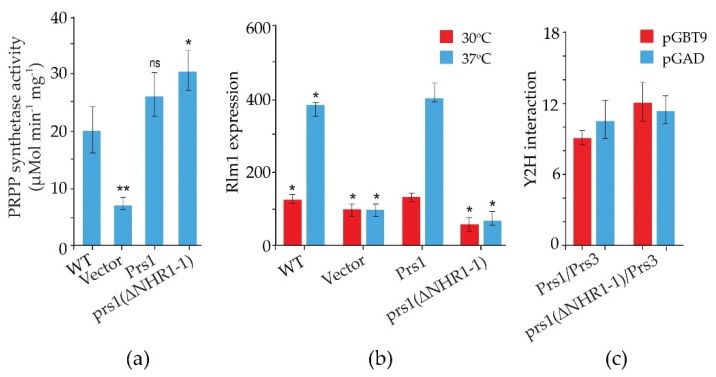
Removal of NHR1-1 from Prs1. (**a**) PRPP synthetase activity in WT or *prs1Δ* strains transformed with the vector, Prs1 or prs1(ΔNHR1-1); (**b**) Rlm1 expression after incubation at 30 °C and 37 °C for the strains shown in (**a**); (**c**) Y2H analysis of the interactions between Prs1/Prs3 and prs1(ΔNHR1-1)/Prs3 in pGBT9 and pGAD vectors carrying the DNA binding or activation domains of the *S. cerevisiae* Gal4 protein. *p*-values: * = *p* ≤ 0.05; ** = *p* ≤ 0.05, ns = not significant.

**Figure 6 cells-11-01909-f006:**
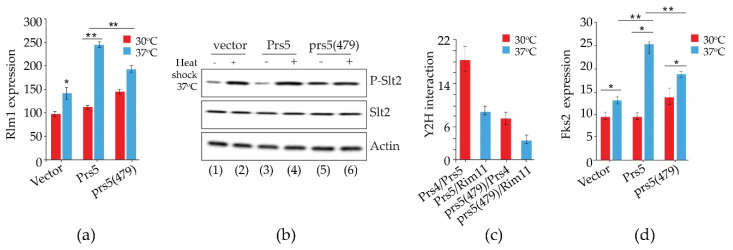
Contribution of the prs5(479) carrying phosphosite mutations, S_364_A S_367_A S_369_A, in NHR5-2 of Prs5: (**a**) Rlm1 expression at the indicated temperatures for *prs5Δ* transformed with vector, Prs5 or prs5(479); (**b**) phosphorylation of Slt2 for the same strains as indicated in (**a**). The cultures were processed for Western blotting, and the membranes challenged with the indicated antibodies shown to the right of the figure in the presence or absence of heat shock. Actin is the loading control. The signals were detected by chemiluminescence. (**c**) Interaction of the yeast Gsk3 kinase, Rim11, with Prs5 and prs5(479). Positive control: Prs4/Prs5; (**d**) Fks2 expression for the same strains under the conditions described in (**a**). * *p* < 0.05, ** *p* < 0.01 (Tukey’s HSD test).

**Figure 7 cells-11-01909-f007:**
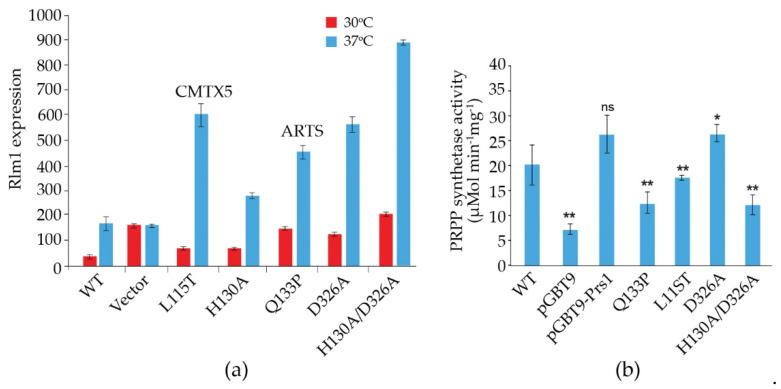
Influence of genocopies p.L115T and p.Q133P in ScPrs1, which correspond to the human neuropathies CMTX5 (p.M115T) and Arts syndrome (p.Q133P), respectively, and characteristic motifs of Prs1 either singly or in combination: H130A, divalent cation-binding site; D326A, PRPP-binding site/Ribose-5-phosphate binding site. It should be noted that the locations of the mutations H130A and D326A differ from those in Figure 2, since the length of Prs1 (427 aa) is increased with respect to Prs2 (318 aa) on account of the insertion of NHR1-1; (**a**) on Rlm1 expression following incubation at 30 °C and 37 °C, all values are significantly different from the WT at *p* < 0.001 (n = 12-20); (**b**) on PRS activity. *p*-values: * = *p* ≤ 0.05; ** = *p* ≤ 0.05, ns = not significant.

**Figure 8 cells-11-01909-f008:**
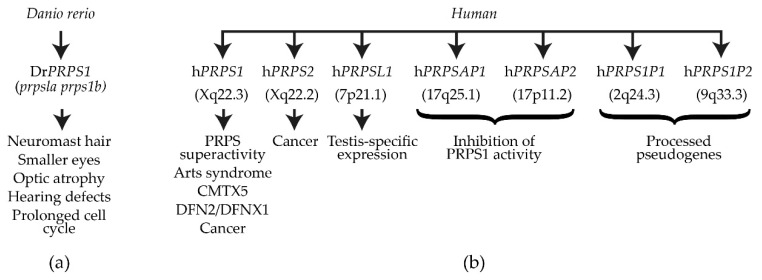
Pleiotropic defects of *PRPS* mutations in zebrafish *Danio rerio* (Dr*PRPS1*) (**a**) and the output of the seven h*PRPS* genes (*PRPS1*, *PRPS2*, *PRPSL1 (PRPS3), PRPSAP1, PRPSAP2, PRPS1P1, PRPS1P2*) (**b**). The chromosomal locations of the h*PRPS* genes are indicated.

**Figure 9 cells-11-01909-f009:**
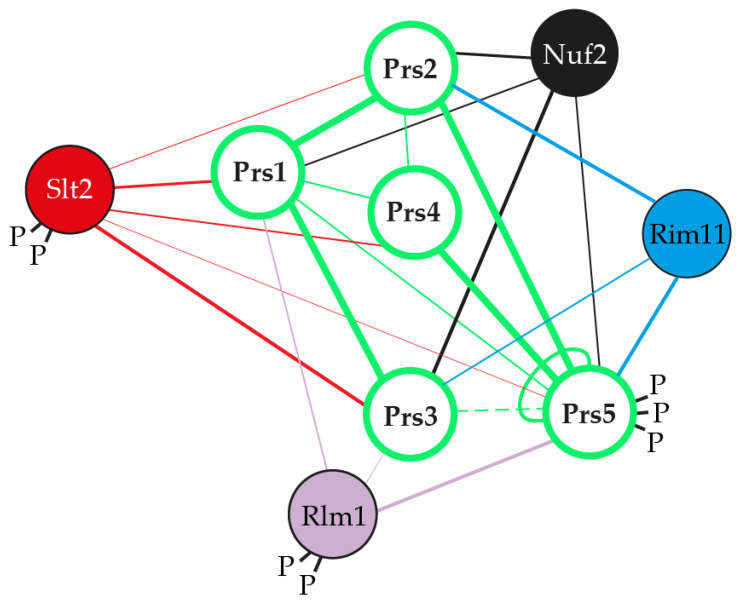
The yeast Prs interactome. The weight of the connectors between the circles representing Prs proteins reflects the strength of the protein/protein interactions between them. The broken line between Prs3 and Prs5 indicates a weak or transient interaction. The heavier the line the stronger the interaction between the Prs polypeptides. The semi-circle on Prs5 [76,232,233] indicates that this protein interacts with itself as do Prs1, Prs2 and Prs3 but not Prs4 [76,232]. For the sake of clarity, the self-interaction is indicated only for Prs5. The remaining connectors indicate the interaction of Prs with Rim11, Rlm1, Slt2 and Nuf2. The last-named is a component of the kinetochore-associated Ndc80 complex localized in the nucleus and involved in chromosome segregation and spindle checkpoint activity. A comprehensive view of all protein/protein interactions in *S. cerevisiae* is available under SGD (https://www.yeastgenome.org, 30 March 2022).

## Data Availability

Not applicable.

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
