# Peer review of "Contribution of Model Organisms to Investigating the Far-Reaching Consequences of PRPP Metabolism on Human Health and Well-Being"

_cells, 2022, doi:10.3390/cells11121909_

Round 1
Reviewer 1 Report
This mauscript summarized the present reasearch status of PRPP metabolism on human health and well-being, and uncover the significance and relevance of human PRPS in disease diagnosis, management and treatment. And this manuscript has certain novelty. I advised that it may be acceptable.
Author Response
This reviewer was very positive and sees the novelty of this review. The input of the reviewer is appreciated.
Reviewer 2 Report
In this review Ugbogu et al focuses on phosphoribosyl pyrophosphate synthetase (PRS) enzyme. PRS is essential for most living organisms as it produces phosphoribosyl pyrophosphate (PRPP), a building block for synthesis of purine and pyrimidine nucleotides, NAD+ and NADP+.
The first part of the review gives a detailed description of PRS genes in Saccharomyces cerevisiae then also mentions PRPS1 paralogues in zebrafish. The second part of the review discusses human PRPS and disorders associated with mutations of PRPS. The topic is highly relevant, both the general understanding of the mechanism of PRPS and the metabolism of PRPP as well as to find potential targets to prevent the symptoms of PRPS-related disorders is important. The cited references are appropriate.
I have two comments regarding the manuscript:
- Even though PRS is expressed in all organisms (except parasites), in this review only Saccharomyces cerevisiae and Danio rerio are discussed as potential model organisms for investigating PRS in regard of human PRS and PRPP metabolism. Mice for example, a more relevant model organism than yeast or zebrafish, should also be mentioned. These models may all provide useful contribution, but there are always limitations. Some of these limitations – for example not complete sequence similarity – may be overcome by the use of cell lines overexpressing human PRPS enzyme. Such cell lines could also provide an ideal tool to screen and test for potential drug targets. It may be worth to discuss the possibility, the advantages and disadvantages of using such cell lines as in vitro models to investigate PRPS and PRPP metabolism.
- The authors briefly mention the limitation of the zebrafish model (lines 502-505), but the limitations of both presented model organisms (yeast and zebrafish) should be discussed in more detail.
A minor spelling comment: make sure that in line 691 “prevalence of less than 1 in 106” 6 is in superscript.
Author Response
Reviewer 2 addresses that mice as model system should also be mentioned. We have accepted his/her advice and included in the relevant sections additional information on the murine system. In the revised ms we have highlighted this in yellow: lines 912-916, 930-944 and 1085-1103.
We agree that overexpression of human PRPS in cell lines would provide an ideal tool to screen for potential drug targets. Specifically, we have included this approach and updated the references accordingly. Please see the following lines 1105-1120.